# Linkage between Dust Cycle and Loess of the Last Glacial Maximum in Europe

Erik J. Schaffernicht[1], Patrick Ludwig[2], and Yaping Shao[1]

[1]Institute for Geophysics and Meteorology, University of Cologne, 50969 Köln, Germany
[2]Institute for Meteorology and Climate Research, Karlsruhe Institute of Technology, 76131 Karlsruhe, Germany

**Correspondence:** Erik J. Schaffernicht (eschaffe@uni-koeln.de)

**Abstract.** This article establishes a linkage between the mineral dust cycle and loess deposits during the Last Glacial Maximum (LGM) in Europe. To this aim, we simulate the LGM dust cycle at high resolution using a regional climate-dust model. The model-simulated dust deposition rates are found to be comparable with the mass accumulation rates of the loess deposits determined from more than 70 sites. In contrast to the present-day prevailing westerlies, winds from northeast, east and southeast (36%) and cyclonic regimes (22%) were found to prevail over central Europe during the LGM. This supports the hypothesis that the recurring east sector winds associated with a high-pressure system over the Eurasian ice sheet (EIS) dominated the dust transport from the EIS margins in eastern and central Europe. The highest dust emission rates in Europe occurred in summer and autumn. Almost all dust was emitted from the zone between the Alps, the Black Sea and the southern EIS margin. Within this zone, the highest emission rates were located near the southernmost EIS margins corresponding to the present-day German-Polish border region. Coherent with the persistent easterlies, westwards running dust plumes resulted in high deposition rates in western Poland, northern Czechia, the Netherlands, the southern North Sea region and on the North German Plain including adjacent regions in central Germany. The agreement between the climate model simulations and the mass accumulation rates of the loess deposits corroborates the proposed LGM dust cycle hypothesis for Europe.

## 1 Introduction

The Last Glacial Maximum (LGM, $21\,000 \pm 3\,000$ yr ago) is a milestone in the Earth's climate, marking the transition from the Pleistocene to the Holocene (Clark et al., 2009; Hughes et al., 2015). During the LGM, Europe was dustier, colder, windier and less vegetated than today (Újvári et al., 2017). The polar front and the westerlies were located at lower latitudes associated with a significant increase in dryness in central and eastern Europe (COHMAP Members, 1988; Peyron et al., 1998; Florineth and Schlüchter, 2000; Laîné et al., 2009; Heyman et al., 2013; Ludwig et al., 2017). The formation of the Eurasian ice sheet (EIS, Fig. 1 and 2) synchronized with a sea level lowering of between $127.5\,\mathrm{m}$ and $135\,\mathrm{m}$ (Yokoyama et al., 2000; Clark and Mix, 2002; Clark et al., 2009; Austermann et al., 2013; Lambeck et al., 2014). It led to different regional circulation patterns over Europe (Ludwig et al., 2016). The greenhouse gas concentrations (185 ppmv $CO_2$, 360 ppbv $CH_4$) were less than half compared to today (Monnin et al., 2001) providing more favorable conditions for $C_4$ than $C_3$ plants. This led to more open vegetation (Prentice and Harrison, 2009; Bartlein et al., 2011) such as grassland, steppe, shrub and herbaceous tundra (Kaplan

et al., 2003; Ugan and Byers, 2007; Gasse et al., 2011; Shao et al., 2018). Central and eastern Europe were partly covered by
taiga, cold steppe or montane woodland containing isolated pockets of temperate trees (Willis and van Andel, 2004; Fitzsimmons et al., 2012). Polar deserts characterized the unglaciated areas in England, Belgium, Denmark, Germany, northern France, western Poland and the Netherlands (Ugan and Byers, 2007). These land surfaces and biome types favored more dust storms and transport over Europe (Újvári et al., 2012).

Loess as a paleoclimate proxy provides one of the most complete continental records for characterizing climate change and evaluating paleoclimate simulations (Singhvi et al., 2001; Haase et al., 2007; Fitzsimmons et al., 2012; Varga et al., 2012). In Europe, loess covers large areas with major deposits centered around 50°N (Antoine et al., 2009b; Sima et al., 2013). However, although numerous European loess sequences date to the LGM, it is not well understood where the dust originated that contributed to the loess formation (Fitzsimmons et al., 2012; Újvári et al., 2017). There are various hypotheses for the potential dust sources, yet they are not fully tested because the dust cycle of the LGM is neither well understood nor quantified. The use of loess as a proxy for paleoclimate reconstruction is considerably compromised because the linkage between the loess deposits and the responsible physical processes is unclear (Újvári et al., 2017). Reliable paleodust modeling is a promising way to establish this linkage and strengthen the physical basis for paleoclimate reconstructions using loess records. Such attempts have been made for example by Antoine et al. (2009b), who analyzed the Nussloch record. They suggested that rapid and cyclic aeolian deposition due to cyclones played a major role in the European loess formation during the LGM.

However, significant discrepancies exist between the mass accumulation rates (MARs) of aeolian deposits that are estimated from fieldwork samples and the dust deposition rates calculated by climate model simulations (Újvári et al., 2010): For Europe, the global LGM simulations result in dust deposition rates (based on different particle size ranges) of less than 100 g m$^{-2}$ yr$^{-1}$ (Werner, 2002; Mahowald et al., 2006; Hopcroft et al., 2015; Sudarchikova et al., 2015; Albani et al., 2016). These are substantially smaller than the MARs (on average: 800 g m$^{-2}$ yr$^{-1}$) that have been reconstructed from more than 70 different loess sites across Europe (Supplementary Table S1). This underestimation is probably due to the coarse resolution of the global models which ignores dust sources, emission, transport and deposition processes at the small scale (Werner, 2002). Other causes can be missing glaciogenic dust sources, a low dust model sensitivity, an underestimated source material availability (Mahowald et al., 2006; Hopcroft et al., 2015), a biased atmospheric circulation, and a lack of dust storms and interannual variability (Hopcroft et al., 2015; Ludwig et al., 2016).

For this study, we simulated the aeolian dust cycle in Europe using a LGM-adapted version of the Weather Research and Forecasting Model coupled with Chemistry (Klose, pers. comm.; Grell et al., 2005; Fast et al., 2006; Kang et al., 2011; Kumar et al., 2014; Su and Fung, 2015) referred to as the WRF-Chem-LGM. The boundary conditions for the WRF-Chem-LGM simulations are provided by the LGM simulation (MPI-LGM) of the Max-Planck-Institute Earth System Model (MPI-ESM; Jungclaus et al., 2012, 2013; Giorgetta et al., 2013; Stevens et al., 2013). This model was chosen since its 1850–2005 experiment reproduces best the recent observed wind distribution over Europe compared to the other climate models (Ludwig et al., 2016). In addition, the MPI-LGM provides three dimensional boundary conditions updated frequently enough to carry out the intended WRF-Chem-LGM experiments. The WRF-Chem was chosen since it has already been evaluated successfully in many recent studies comparing its dust simulations with observations (Bian et al., 2011; Kang et al., 2011; Zhao et al.,

2011, 2012; Rizza et al., 2016; Baumann-Stanzer et al., 2019). Therefore, it is likely that the newly created WRF-Chem-LGM will simulate the LGM dust emission, transport and deposition processes similarly well. This capacity of the WRF-Chem-
LGM allows reducing the discrepancies between the MARs and the simulation-based dust deposition rates. It enables the establishment of a linkage between the glacial dust cycle and the on site loess deposits.

## 2  Data and Methods

The WRF-Chem-LGM consists of fully coupled modules for the atmosphere, land surface, and air chemistry. The simulation

domain encompasses the European continent including western Russia and most of the Mediterranean (Fig. 1) discretized by a grid spacing of 50 km and 35 atmospheric layers. The domain boundary conditions were 6-hourly updated by using the
MPI-LGM. The sea surface temperature and sea ice cover are updated daily based on the corresponding MPI-LGM variables. To simulate the dust cycle including dust emission, transport and deposition, the dust-only mode of the WRF-Chem-LGM was
selected. This mode implies the application of the size-resolved (dust size bins: 0–2, 2–3.6, 3.6–6, 6–12 and 12–20 μm) University of Cologne dust emission scheme (Shao, 2004), the Global Ozone Chemistry Aerosol Radiation Transport (GOCART;
Chin et al., 2000; Ginoux et al., 2001; Chin et al., 2002; Ginoux et al., 2004), the dry and the wet deposition modules (Wesely, 1989; Chin et al., 2002; Grell et al., 2005; Jung et al., 2005).
To replace the present-day WRF surface boundary conditions by the LGM conditions, the data sets for the global 1° resolved land-sea mask and the topography offset provided by PMIP3 (Paleoclimate Model Intercomparison Project Phase 3; Braconnot
et al., 2012) were interpolated to the 50 km grid (Fig. 1, Supplementary Table S2 and S3). To represent the LGM glaciers and land use, the 2° CLIMAP reconstructions (Climate: Long range Investigation, Mapping, and Prediction; Cline et al., 1984)
were also interpolated to the 50 km grid and converted (Ludwig et al., 2017) to the WRF-compatible United States Geological Survey categories (USGS-24) to replace their present-day analogs. The relative vegetation seasonality during the LGM is
assumed to resemble to the present. Based on this uniformitarianism approach, the CLIMAP maximum LGM vegetation cover reconstruction (Cline et al., 1984) was weighted using the corresponding monthly fractions of the present-day WRF maximum
vegetation cover and prescribed in the model.

The erodibility at point $p$ during the LGM is approximated by

$$S = \left( \frac{z_{\max} - z}{z_{\max} - z_{\min}} \right)^5 \tag{1}$$

with $z$ being the LGM terrain height at $p$ and $z_{\min}$ ($z_{\max}$) representing the minimal (maximal) height in the $10° \times 10°$ area

centered around $p$ (Ginoux et al., 2001). Setting $S$ to zero where the CLIMAP bare soil fraction reconstruction is less than 0.5 refines this approximation. The adapted University of Cologne dust emissions scheme takes into account that the erodi-
bility exceeds a lower limit of 0.09 for emission to occur. This suppresses dust sources in areas that had been attributed small physically meaningless interpolation-caused erodibility artifacts. The vegetation and snow cover are considered mutually in-
dependent and uniformly distributed within a grid cell, i.e. the erodible area is multiplied by the fractional factor $(1 - c_{\text{snow}})$ to account for snow cover.
To simulate the LGM dust cycle with the WRF-Chem-LGM, two downscaling approaches of the MPI-LGM were implemented: the dynamic downscaling approach and the statistic dynamic downscaling approach. Both emerge from simulations
that base on identically configured numerical schemes representing the atmospheric chemistry and physics in the WRF-Chem-LGM. Using dynamic downscaling, a consecutive 30 year simulation (corresponding to more than 10 000 days) was performed.
In contrast, the statistic dynamic downscaling is based on 130 mutually independent episodes each spanning eight days, or a total of 1040 days. The episode selection relies on the Circulation Weather Type (CWT) classification (Jones et al., 1993, 2013;
Reyers et al., 2014; Ludwig et al., 2016) of the MPI-LGM records into ten classes: Cyclonic, Anticyclonic, Northeast, East, Southeast, South, Southwest, West, Northwest and North. The CWT classification approach is chosen since the atmospheric
circulation patterns are the dominant factor for controlling dust emission from and deposition on dry, low and sparsely vegetated soil surfaces (Ginoux et al., 2001; Darmenova et al., 2009; Shao et al., 2011a, b). Such kind of surfaces characterized the
unglaciated regions in central and eastern Europe during the LGM (Ugan and Byers, 2007). To compare the prevailing wind directions over Europe during the Pre-Industrial (PI) and the LGM, the daily mean sea level pressure patterns (interpolated to
2.5° horizontal grid spacing) of the MPI-LGM and the MPI-ESM simulation for the PI (MPI-PI) were classified for the region centering around (17.5°E, 47.5°N). For records showing rotational and directional CWT patterns, only the directional pattern
is counted. By counting and statistically evaluating the CWTs of all records, a LGM and a PI CWT occurrence frequency distribution is established. The LGM distribution served to reconstruct the LGM dust cycle using statistic dynamic downscaling.
It also enabled analyzing the contributions of each wind regime to the dust cycle.

For the statistic dynamic downscaling, we performed 130 WRF-Chem-LGM simulations in total, i.e. 13 simulations for

each of the 10 CWT classes. For each of these eight-day spanning simulations, independent consecutive sequences of boundary conditions were chosen out of all MPI-LGM records of the same CWT class. For CWTs with too few sets of distinct consecutive
MPI-LGM records of the required CWT, the remaining sets were chosen applying less strict selection criteria (Table 1). For the analysis of all performed episodic simulations, the first two days of each episode are considered as spin-up days and excluded.
The reconstruction of quantity $Q$ using statistic dynamic downscaling is then calculated from the weighted ensemble mean (Reyers et al., 2014):
$$\langle Q \rangle = \sum_i \frac{f_i}{T} \int_T Q(t)dt \tag{2}$$

with $i$ representing the $i^{\text{th}}$ CWT, $f_i$ its occurrence frequency and $T$ its duration. To evaluate the simulations, the obtained dust

deposition rates are compared to more than 70 independent MARs reconstructed from loess sites located in the simulation domain (Supplementary Table S1).

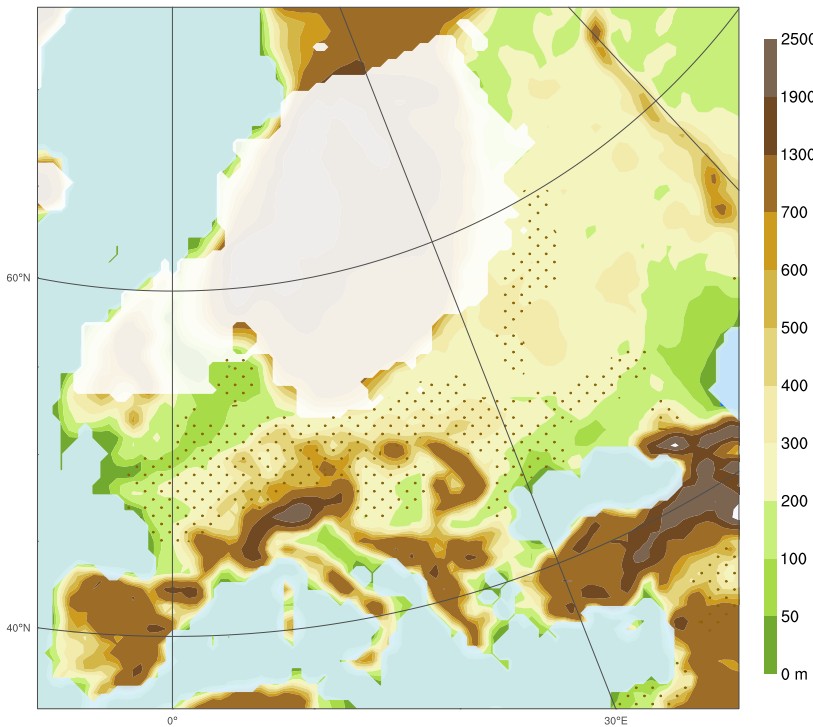

**Figure 1.** Simulation domain showing the applied topography (shaded), the potential dust source areas (dots) and the Eurasian ice sheet extent (white overlay, adapted from Cline et al., 1984) of the Last Glacial Maximum.

**Table 1.** Temporal concept for the episodic eight-day WRF-Chem-LGM simulations performed to reconstruct the LGM dust cycle based on statistic dynamic downscaling. As the MPI-LGM contains for a few CWTs less than 13 separate eight-day record sequences, some of the episodes were driven by a heterogeneous sequence of records. That is, one (or more) of the records in these sequences differs in its CWT from the CWT of the records for the main days. For selecting heterogeneous sequences, the CWT-correspondence between the main and tracking records is considered of higher priority (= [++]) than between main and spin-up records (= [+]).

|  | Days | Preferences for selecting record series from the MPI-LGM |
|---|---|---|
| Spin-up | 2 | Prefer[+] sequences whose spin-up records have the same CWT as the main records |
| ↓ | | |
| Main | 3 | All records that drive the main part (central 3 days) of each episode must be of the same CWT |
| ↓ | | |
| Tracking | 3 | Prefer[++] sequences whose tracking records have the same CWT as the main records |

# 3 Results

## 3.1 Dust Cycle Hypothesis

In line with previous modeling (COHMAP Members, 1988; Ludwig et al., 2016) and fieldwork studies (Dietrich and Seelos, 2010; Krauß et al., 2016; Römer et al., 2016), we hypothesize that east sector winds (i.e. northeasters, easterlies and southeasters) dominated the mineral dust cycle over central Europe during the LGM (Fig. 2). This hypothesis also implies a linkage of dust sources in central and eastern Europe during the LGM and the loess deposits in Europe. It is suggested here that a greater proportion of all LGM dust deposits in central and eastern Europe comes more from sources in central and eastern Europe than from sources in the Channel. The east sector winds likely contributed substantially to the formation of the European loess belt in central Europe. Among them, the northeasters and easterlies originated most likely from dry winds that flowed down the slopes of the southern and eastern EIS margins where they picked up and turned gradually into northeasters and easterlies. By blowing over the bare proglacial EIS areas, they generated dust emissions, carried the dust westwards implying dust depositions in areas west of the respective dust sources.

## 3.2 East Sector Winds and Cyclones over Central Europe

In agreement with this hypothesis, glacial simulations for 90 ka ago evidenced katabatic winds over the EIS (Krinner et al., 2004) and GCM simulations for the LGM indicate prevailing east sector winds over central and eastern Europe (COHMAP Members, 1988; Ludwig et al., 2016). In Germany, several aeolian sediment records that are dated to the LGM originated from more eastern sources (Dietrich and Seelos, 2010; Krauß et al., 2016; Römer et al., 2016). The CWT frequencies for the present (not shown) and the PI are very similar, therefore it is possible to use the term present-day to refer to both the PI and the actual present-day frequencies. In contrast to the dominant present-day anticyclones and west sector winds (southwesters, westerlies and northwesters), east sector winds (36%) and cyclones (22%) prevailed over central Europe during the LGM (Table 2). The east sector winds are associated with a strong EIS-High (Fig. 2a and COHMAP Members, 1988). The increased frequency of cyclones over central Europe is consistent with the analysis of the LGM storm tracks, which deviated from their present-day course (Hofer et al., 2012), running either along central Europe, the Mediterranean or the Nordic Seas (Florineth and Schlüchter, 2000; Luetscher et al., 2015; Ludwig et al., 2016). Their Mediterranean course is consistent with the Alpine, western, and southern European climate proxies (Luetscher et al., 2015). In addition, the proxies indicate a storm track branch split-off over the Adriatic that ran past the Eastern Alps to central Europe (Florineth and Schlüchter, 2000; Luetscher et al., 2015; Újvári et al., 2017). These proxy-based findings are in line with the more frequent cyclones in central Europe during the LGM (Table 2). This, in turn, can be related to the stronger and southwards shifted jet stream (Luetscher et al., 2015; Ludwig et al., 2016) and the missing Scandinavian cyclone tracks, which were deflected southwards by the blocking EIS-High. As a result, their frequency increased over central Europe (Table 2), consistent with susceptibility- and grain-size-based results that suggest more frequent storms over western Europe. The east sector winds, which more than doubled in frequency in comparison to today (36% compared to 17%, Table 2) need to be incorporated to establish a more complete understanding of the main drivers of the dust cycle in Europe during the LGM (Fig. 9a). These winds are also evidenced by northern-central

European grain-size records for the Late Pleniglacial (Bokhorst et al., 2011). Sediment layers attributed to east wind dated to 36–18 ka BP are abundant in the Dehner Maar sediments (Eifel, Germany, 6.5°E, 50.3°N; Dietrich and Seelos, 2010). Their provenance showed that up to every fifth dust storm over the Eifel came from the east (Dietrich and Seelos, 2010).

Our findings are in agreement with fieldwork-based results of Römer et al. (2016), who found evidence for strong east sector winds over northern, central and western Germany for 23 to 20 ka ago. Also loess in the Harz Foreland indicates a shift to prevailing east sector winds for the LGM (Krauß et al., 2016). The location of aeolian ridges along rivers in northeastern Belgium and a core transect near Leuven also support our finding by evidencing northeasters for the Late Pleniglacial (Renssen et al., 2007). In addition, northerlies, northeasters and easterlies were inferred from loess deposits west of the Maas (Renssen et al., 2007). Also for Denmark, wind-polished boulders evidence dominant easterlies and southeasters in the period of 22 to 17 ka ago (Renssen et al., 2007). The CWT frequency distribution for the LGM (Table 2) contradicts the finding (Renssen et al., 2007) of prevailing west sector winds during the LGM in central Europe (0–30°E, 40–55°N). The distribution also contrasts with the finding (Sima et al., 2013) of prevailing winds from west-northwest in eastern central Europe, in particular for the area around Stayky (31°E, 50°N). More precisely, the CWT-W and CWT-NW regimes occurred in eastern central Europe in sum for less than 10% of the times during the LGM (Table 2), which is even less than the expectation value for a single weather type in case of a uniform CWT frequency distribution. On the contrary, the significant role of the east sector winds (Table 2) is consistent with the deposits on the west bank of the Dnieper (Sima et al., 2013), which are also the loess deposits closest to Stayky. In addition, sandy soil texture and sand dunes indicate prevailing northerlies and northeasters over Dobrudja (28.18°E, 44.32°N), the eastern Walachian Plain (both located in Romania) and Stary Kaydaky (Ukraine, 35.12°E, 48.37°N; Buggle et al., 2008). The northerlies over Ukraine originated from katabatic winds descending from the EIS (Buggle et al., 2008). The high aridity and grain size variations of the Surduk and Stari Bezradychy records (Serbia/Ukraine, Supplementary Table S1) evidence prevailing dry and periodically strong east sector winds (Antoine et al., 2009a; Bokhorst et al., 2011).

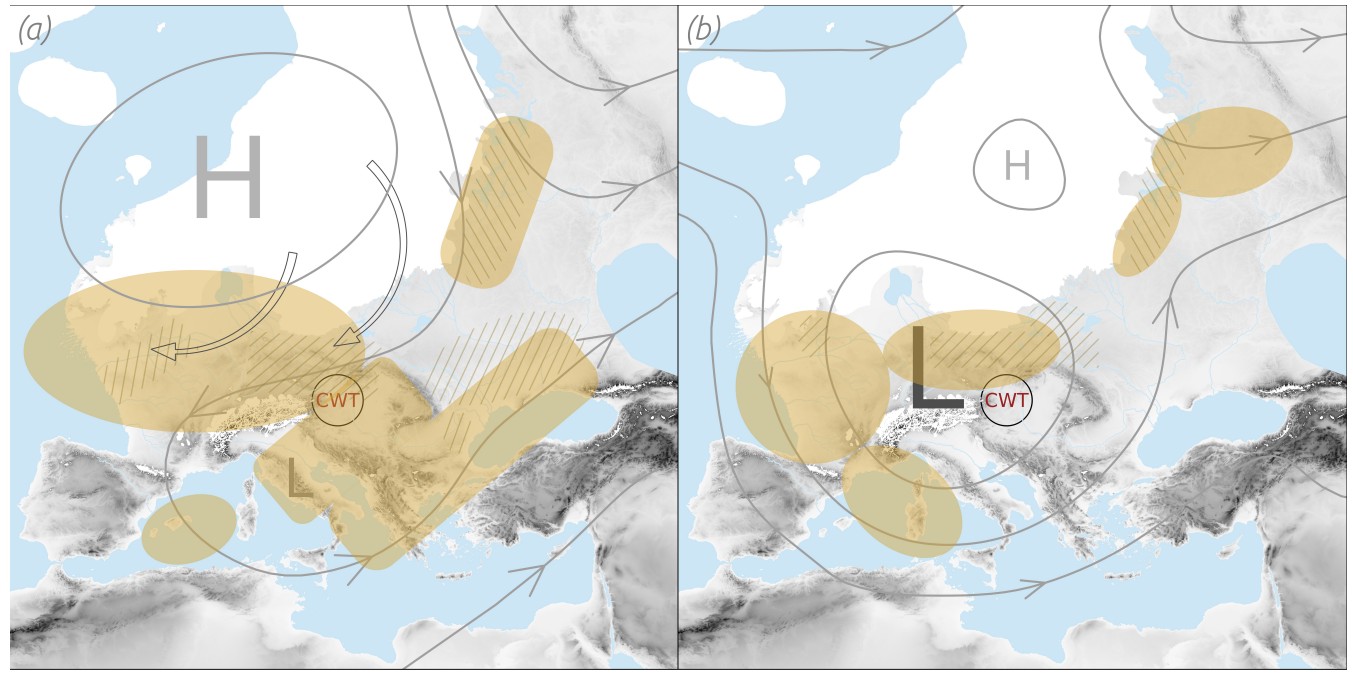

**Figure 2.** Conceptual model explaining the linkage between the European dust cycle during the Last Glacial Maximum and the loess deposits. The main dust deposition areas (filled), emission areas (hatched), wind (grey lines) and pressure patterns (H/L: high/low pressure) are highlighted; all of them result from the WRF-Chem-LGM experiments. The center of the region for the Circulation Weather Type analysis is denoted with CWT. (a) Northeasters, easterlies and southeasters (the east sector winds; transparent arrows with black perimeter) caused by the semi-permanent high-pressure over the Eurasian ice sheet (white) prevailed 36% of the time over central Europe (Table 2). (b) The cyclonic weather type regimes which prevailed 22% of the time over central Europe (Table 2).

**Table 2.** Circulation Weather Type occurrence frequencies (%) for central Europe (centered at 17.5°E and 47.5°N) during the LGM and the Pre-Industrial period (PI). The frequencies are based on the LGM and the PI simulation of the Max-Planck-Institute Earth System Model. The Circulation Weather Type classes are: Cyclonic (C), Anticyclonic (A), Northeast (NE), East (E) followed by the remaining standard wind directions.

|     | C | A | NE | E | SE | S | SW | W | NW | N |
|-----|------|------|------|------|------|-----|------|------|-----|-----|
| LGM | 22.2 | 8.9 | 12.4 | 13.4 | 10.2 | 9.7 | 6.8 | 4.3 | 5.0 | 7.0 |
| PI  | 10.6 | 24.1 | 7.3 | 5.2 | 4.9 | 7.6 | 11.6 | 11.1 | 9.4 | 8.3 |

## 3.3 Dust Emissions from the Eurasian ice sheet margin

The model-simulated dust emission (Fig. 3) indicates that most dust in Europe was emitted from the less elevated corridor between the Alps, the Black Sea and the EIS (45–55°N). This finding is consistent with loess-based dust-flux estimates (Újvári et al., 2010). The highest emission rates ($>10^5$ g m$^{-2}$ yr$^{-1}$) occurred along the southern EIS margin (15–18°E, 51–53°N, Fig. 3). This location is in line with the location of the highest emissions found in the Greenland stadial GCM simulation of Sima et al. (2013), yet, our simulation indicates a larger upper limit for the emission rates (1000 g m$^{-2}$ yr$^{-1}$). Our results also show high emissions in the dry-fallen Channel and the German Bight (Fig. 3). For the latter, they compare well with a glacial climate simulation that calculated an average emission of 140 and a maximal emission of $>200$ g m$^{-2}$ yr$^{-1}$ (Sima et al., 2009).

The loess deposits (Újvári et al., 2010) and the model results are consistent in that the Carpathian Basin was both a dust source and a dust sink (Fig. 3 and 4). Major dust sources surrounding the Carpathians and the Eastern Alps (Fig. 3) are in line with deposits in Serbia and the Carpathian Basin (Újvári et al., 2010; Bokhorst et al., 2011). The dust emissions from the Lower Danube Basin (Fig. 3) are in agreement with plentiful sediment supply, strong winds and dry conditions inferred from the plateau loess in Urluia, located near the Black Sea in southeastern Romania (Fitzsimmons and Hambach, 2014). Also the emissions from the western Black Sea littoral (Fig. 3) are consistent with provenance analyses of Eastern Dobrogea loess in the Lower Danube Basin (Jipa, 2014). Our results indicate a close relationship between strong dust emissions and low terrains (or basins). This relationship is found for the North Sea Basin and the European plains bordering the EIS, the Caucasus, the Carpathians or the Massif Central (Fig. 1 and 3). The dust emissions from the EIS margin and from the foothills of the European mountains (Fig. 3) are consistent with the loess-based finding of significant aeolian dust contributions from glaciogenic and orogenic dust sources (Újvári et al., 2010).

## 3.4 Conforming Dust Deposition and Loess Accumulation Rates

Compared with the GCMs (Werner, 2002; Mahowald et al., 2006; Hopcroft et al., 2015; Sudarchikova et al., 2015; Albani et al., 2016), the WRF-Chem-LGM dust deposition rates ($F_D$, Fig. 4) reproduce the MARs (Supplementary Table S1, Fig. 4a and b) and MAR10 (Supplementary Table S1, Fig. 4c and d) better, at least by one order of magnitude. One factor for this improvement is most likely the higher spatiotemporal resolution (Ludwig et al., 2019) of the WRF-Chem-LGM experiments combined with the provided higher resolved geographical input data, for example the regional LGM topography, land use and dynamic (yet monthly prescribed) vegetation cover. The boundary conditions provided by the MPI-LGM could also be a factor for this improvement. Taking into account that the MPI-ESM experiment for the present reproduces the observed atmospheric circulation over Europe better than other GCMs (Ludwig et al., 2016), it is likely that MPI-LGM also reproduces the LGM conditions more realistically. Another factor could be the orography-based estimated fraction of alluvium (Ginoux et al., 2001) combined with the proxy-based reconstructed bare soil fraction (Cline et al., 1984) to calculate the spatial erodibility distribution. Based on this distribution, the WRF-Chem-LGM was able to suppress unrealistic numerical dust emission from areas with low or zero erodibility. Most likely, the improvement results also from selecting the well-tested and observation-confirmed Shao dust emission scheme (Shao, 2004; Kang et al., 2011). For example, this scheme takes into account the dynamic

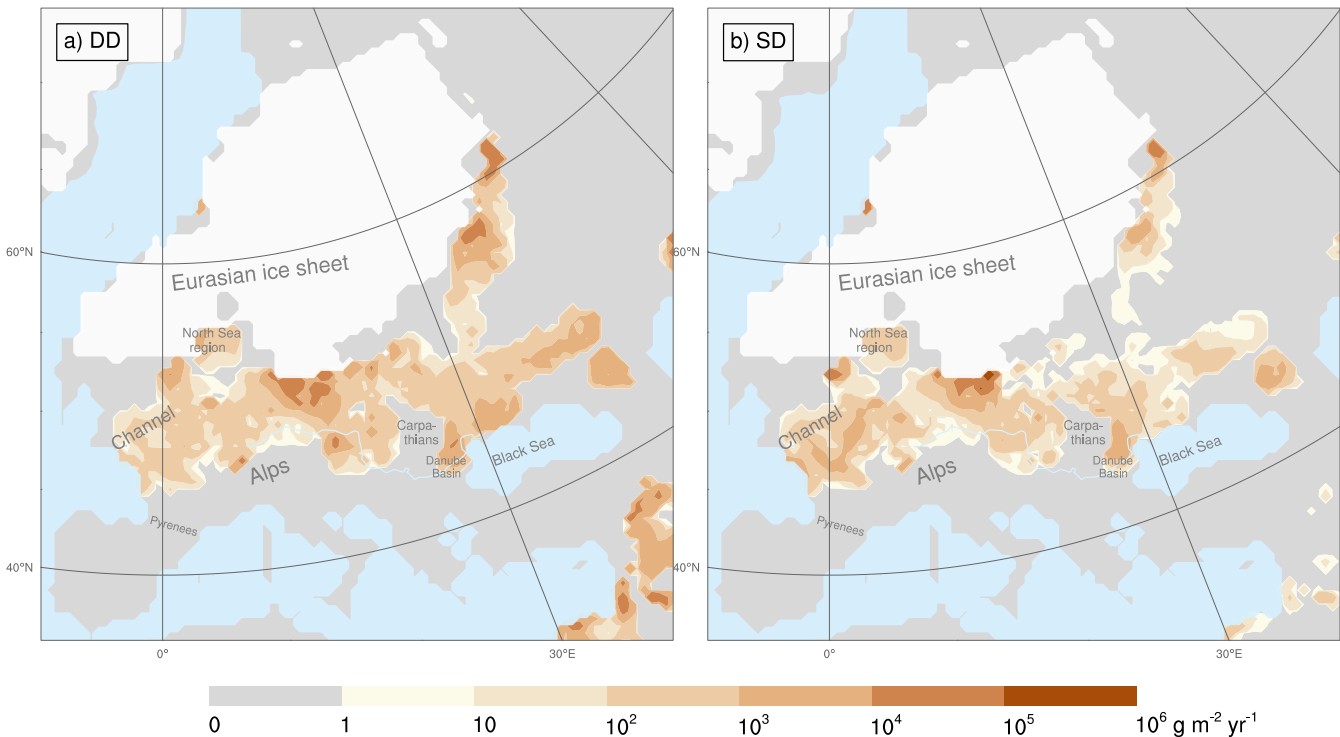

**Figure 3.** Dust emission rates for the Last Glacial Maximum. These reconstructions are based on a) dynamic downscaling (DD) and b) statistic dynamic downscaling (SD). Ice sheet extents (white overlay), Danube (light-blue line).

moisture changes at the soil surface. Due to our recent improvement of the Shao dust emission scheme, the effect of snow cover
on dust emission has also been taken into account in the WRF-Chem-LGM experiments.

The MARs and MAR10 (Supplementary Table S1 and Fig. 4) were reconstructed from samples that were extracted during
fieldwork campaigns from loess paleosol sites. The MAR for a specific site was inferred by taking into account all particles
found in respective sample, independent of their diameter. In contrast, the MAR10 for the same site was inferred by taking into
account only particles up to 10 μm diameter. Most of the MARs and MAR10 (Supplementary Table S1 and Fig. 4) result from
sites of the European loess belt. This belt plays a key role in assessing paleoclimatic dust cycle simulations for Europe (Kukla,
1977; Little et al., 2002; Haase et al., 2007; Sima et al., 2009). During the LGM, it corresponded approximately to the fraction of
the European land area that was bounded northwards by the EIS and southwards by the Alps, Dinaric Alps and Black Sea. The
$F_{D20}$ in Figure 4a and b ($F_{D12}$ in Fig. 4c and d) denote the WRF-Chem-LGM deposition rates caused only by particles smaller
than 20 μm (12 μm) in diameter. To distinguish the deposition rates obtained from the two downscaling methods, the $F_{D20\,DD}$
and $F_{D12\,DD}$ relate to the dynamic, while the $F_{D20\,SD}$ and $F_{D12\,SD}$ relate to the statistic dynamic downscaling simulations. For
central Europe, the dynamic (Fig. 4a and c) and statistic dynamic downscaling (Fig. 4b and d) resulted in similar $F_D$ values
confirming the suitability of the statistic dynamic downscaling.

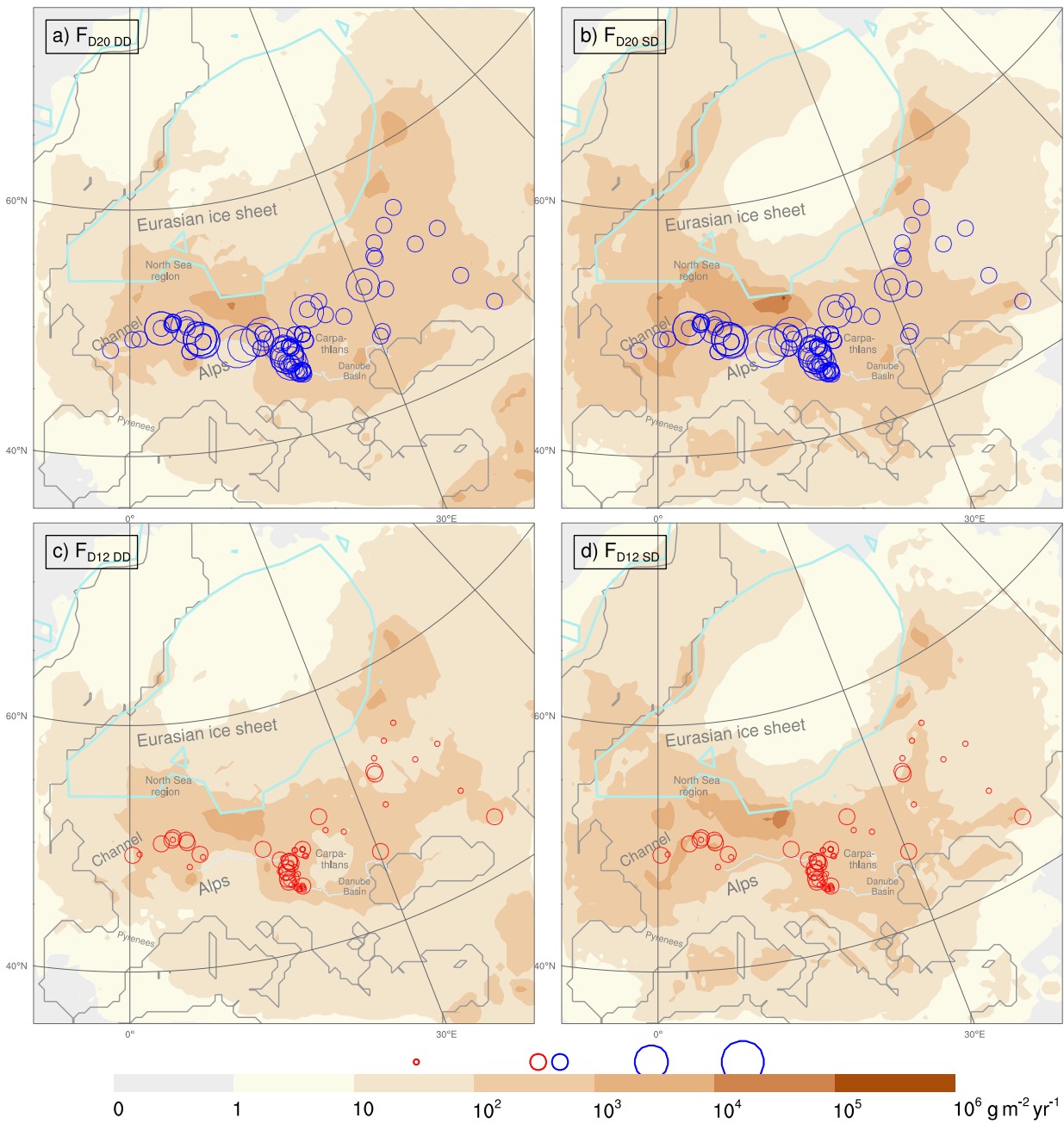

**Figure 4.** Dust deposition rates for the Last Glacial Maximum, comprising particles of up to 20 μm diameter ($F_{D20}$) using (a) dynamic downscaling ($F_{D20\,DD}$) and (b) statistic dynamic downscaling ($F_{D20\,SD}$). (c) and (d) as (a) and (b), but for particles up to 12 μm ($F_{D12}$). Each blue circle size represents one mass accumulation rate (MAR, Supplementary Table S1 column 5) magnitude. Each red circle size represents one reduced mass accumulation rate (MAR10, Supplementary Table S1 column 6) magnitude. MAR and MAR10 values compiled in Supplementary Table S1. The simulation-based ($F_{D20}$, $F_{D12}$) and the fieldwork-based (MAR, MAR10) rates result from independent data. Delineated are the Danube (light blue), the coastlines (grey; Braconnot et al., 2012) and the ice sheet extents (turquoise; Cline et al., 1984).

During the LGM, the largest $F_{D20}$ ($>10^5$ g m$^{-2}$ yr$^{-1}$) occurred in western Poland (Fig. 4a). Slightly lower $F_{D20}$ ($10^4$–$10^5$ g m$^{-2}$ yr$^{-1}$) were found in adjacent areas, e.g. in eastern Germany. $F_{D20}$ was $10^3$–$10^4$ g m$^{-2}$ yr$^{-1}$ on the North German Plain, in the dry-fallen German Bight, eastern England, northern and western France, the Benelux and southeast of the Carpathians. Regional deposition maxima of $10^3$–$10^4$ g m$^{-2}$ yr$^{-1}$ occurred along the French LGM coastline (46–48°N), on the eastern side of the Carpathians (44–47°N, including the eastern Romanian Danube Plain) and near the Caucasus (44–45°N, Fig. 4a). They coincide with today's extensive loess derivates along the Atlantic coastline of France, at the European foothills north of 42°N and with the loess thickness maximum in the Romanian Danube Plain (Haase et al., 2007; Jipa, 2014). The quality of the simulations is also recognizable in the Carpathian Basin, which is now half covered with loess and clay of aeolian origin (Varga et al., 2012). There, the simulated $F_{D20}$ of 100–1000 g m$^{-2}$ yr$^{-1}$ (Fig. 4a) are in good agreement with the MARs (200–500 g m$^{-2}$ yr$^{-1}$). In Ukraine and at the eastern margins of the EIS, $F_{D20}$ of 100–1000 g m$^{-2}$ yr$^{-1}$ are in line with the MARs (Fig. 4a). Over Ukraine and consistent with our results, dust transport and deposition by east sector winds is evidenced by loess deposits on the west bank of the Dnieper (Sima et al., 2013).

The MARs of a few loess sites are higher than the $F_{D20}$ in their surrounding. Such an underestimation could be explained by particles larger than 20 μm which are not taken into account by the $F_{D20}$. For some regions, the MARs of closely related sites vary over orders of magnitude, e.g. between $10^2$ and $10^4$ g m$^{-2}$ yr$^{-1}$ near the Rhine and in Belgium (Fig. 4a). This may be due to strong small scale variability, loess dating uncertainties (Singhvi et al., 2001; Renssen et al., 2007) or age model inaccuracies (Bettis et al., 2003). For western Germany, a transition from higher $F_{D20}$ ($10^3$–$10^4$ g m$^{-2}$ yr$^{-1}$) in its northeast to lower $F_{D20}$ ($10^2$–$10^3$ g m$^{-2}$ yr$^{-1}$) in its southwest was found (Fig. 4a). For a few sites in southwestern Germany, Austria, Ukraine and along the Danube, $F_{D20}$ is an order of magnitude lower than the respective MARs (Fig. 4a). Given the 50 km grid spacing of the WRF-Chem-LGM simulation, this may be attributed to missing local dust sources, such as dry-fallen riverbeds and floodplains. Possibly, the MARs of these sites are also inferred from particles that were predominantly larger than 20 μm yet data on particle sizes is not available. The peak deposition locations and the overall shape of the $F_{D20}$ and $F_{D12}$ patterns are very similar (Fig. 4). The $F_{D12}$ are also almost everywhere consistent with the MAR10 (Fig. 4c and d). Those $F_{D12}$ that overestimate the MAR10 do not contradict the consistency since the $F_{D12}$ also take into account particles that are (by definition) excluded by the MAR10. In summary, large consistency was found between the simulated dust deposition rates and the MARs and MAR10 that were reconstructed from on-site samples.

### 3.5 Seasonal dust cycle patterns

During the LGM, the strongest emission and deposition in Europe occurred in summer, followed by autumn and spring (Fig. 5 and 6). The areas with the overall highest emission were also those with the highest seasonal emission (Fig. 3 and 5). The spring and winter emissions have the same order of magnitude. The low winter and spring emission rates along the EIS margin were caused by the then extensive snow cover there. During winter, emissions peaked only in northern France, consistent with its small snow cover and the vegetation cover (Fig. 7) that was prescribed to the WRF-Chem-LGM. Major dust emissions occurred from the Carpathian Basin and along the northwest coast of the Black Sea. During spring, slightly attenuated emissions are simulated for France, despite of the decreasing snow cover but in accordance with its increasing vegetation cover. Consid-

erably higher emission rates are simulated from along the German and Polish EIS margin where the snow cover had retreated. For eastern Europe, the growing vegetation cover and the slight soil moisture increase account for partly lower spring than winter emission rates. The soil moisture increase possibly resulted from meltwater of the retreating snow cover. The highest emission rates occurred during summer and were located along the German and Polish EIS margin. Slightly lower emissions are found to the east of the EIS. These findings are coherent with the surface properties of these areas during summer, i.e. they were mostly snow-free and the least moist. During fall, the snow cover increased, causing a decrease of dust emissions, except for the area north of the Black Sea which encountered its annual maximum. This maximum can be attributed to the retreat of the vegetation cover and the dry soil conditions there.

The winter CWT distribution indicates prevailing east sector winds (37%) in contrast to cyclonic regimes, which occurred much less frequently than on annual average (13%; Table 2 and 3). The winter deposition rates northwest of the Alps were considerably above, while the rates at the central and eastern European EIS margin were below the annual average (Fig. 4 and 6a). In western Europe, the highest deposition rates occurred near the sources, yet a considerable dust fraction was also transported and deposited to the west and northwest of the sources, which requires east sector winds. Low deposition rates were found for southern France, however marked depositions occurred when subjected to cyclonic regimes (Fig. 9b). The deposition pattern for the central Mediterranean area (Italy, the Adriatic) suggests significant dust transport by east sector winds and anticyclonic winds, in sum prevailing 51% of the times. In eastern Europe, considerable winter depositions rates covered areas south of the dust sources, in particular the western Black Sea and regions south of the Danube. This indicates a significant contribution to the dust transport by northerlies (6%), northeasters (12%) and the anticyclonic regimes (14%).

Also the spring deposition rates evidence the importance of the east sector winds (42%, Table 3) for the dust cycle. In western Europe, major deposition areas are to the west and northwest of the sources, while they are to the west and southwest in eastern Europe (Fig. 6b). An increase of the dust transport towards the south in western, and towards the north in eastern Europe indicates an increasing role of the cyclonic regimes (27%) during the spring.

The summer deposition rates are distributed zonally along the EIS margin, suggesting an approximately latitude-parallel dust transport by west (21%) and/or east sector (24%) wind directions. In addition, the northern flanks of cyclonic regimes (24%) likely contributed to a westwards dust transport. Over north-easternmost Europe (40E, 62N), the deposition rates suggest east sector winds. The autumn deposition rates over western and central Europe show a westward running plume from the southern EIS margin over Germany and Poland, corroborating the major role of the east sector winds (38%) for the dust cycle. The high deposition rates in eastern Europe suggest that also the cyclonic regimes (19%) contributed during fall.

**Table 3.** Seasonal CWT occurrence frequencies (%) for central Europe (centered at 17.5°E and 47.5°N) during the LGM. The frequencies are based on MPI-LGM simulation. The CWT classes are: Cyclonic (C), Anticyclonic (A), Northeast (NE), East (E) followed by the remaining standard wind directions. Sum E is the sum of the east sector winds (NE, E, SE). The seasons are labeled DJF (winter), MAM (spring), JJA (summer) and SON (fall).

|     | C    | A    | Sum E | NE   | E    | SE   | S    | SW  | W   | NW  | N    |
|-----|------|------|-------|------|------|------|------|-----|-----|-----|------|
| DJF | 12.6 | 13.9 | 37.4  | 11.8 | 14.4 | 11.2 | 12.9 | 8.5 | 5.1 | 4.1 | 5.6  |
| MAM | 27.1 | 6.1  | 41.9  | 12.9 | 16.4 | 12.6 | 9.7  | 4.8 | 2.8 | 3.6 | 4.2  |
| JJA | 26.8 | 7.5  | 24.4  | 12.8 | 6.3  | 5.3  | 9.3  | 7.3 | 6.1 | 7.9 | 10.7 |
| SON | 18.6 | 10.0 | 37.8  | 12.8 | 13.6 | 11.4 | 10.8 | 6.8 | 3.8 | 5.1 | 7.0  |

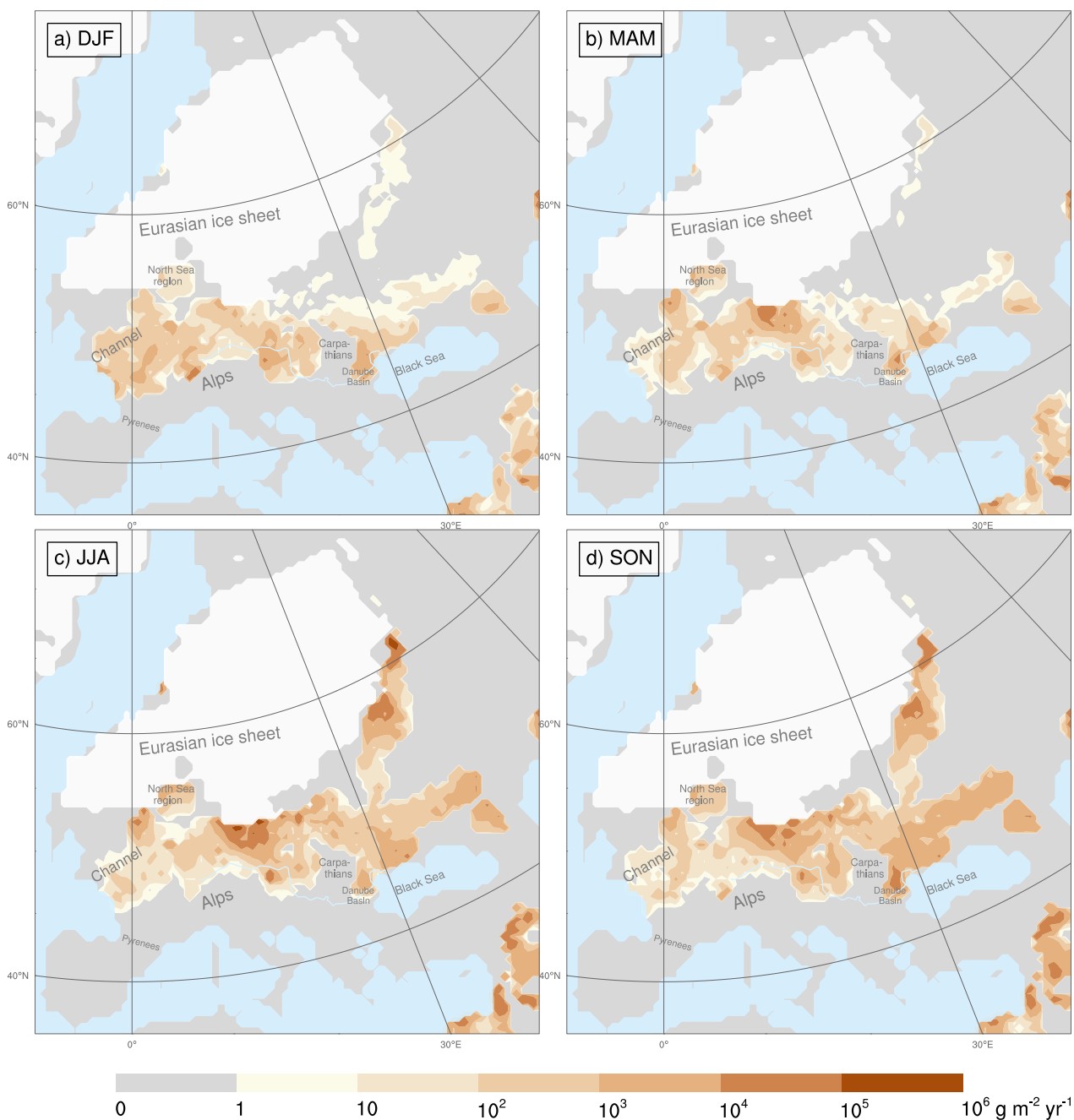

**Figure 5.** Dust emission rates for a) winter (DJF), b) spring (MAM), c) summer (JJA), and d) fall (SON) during the Last Glacial Maximum. This reconstruction is based on dynamic downscaling. The Danube (light-blue line) and the extent of the continental ice sheets (white) are shown.

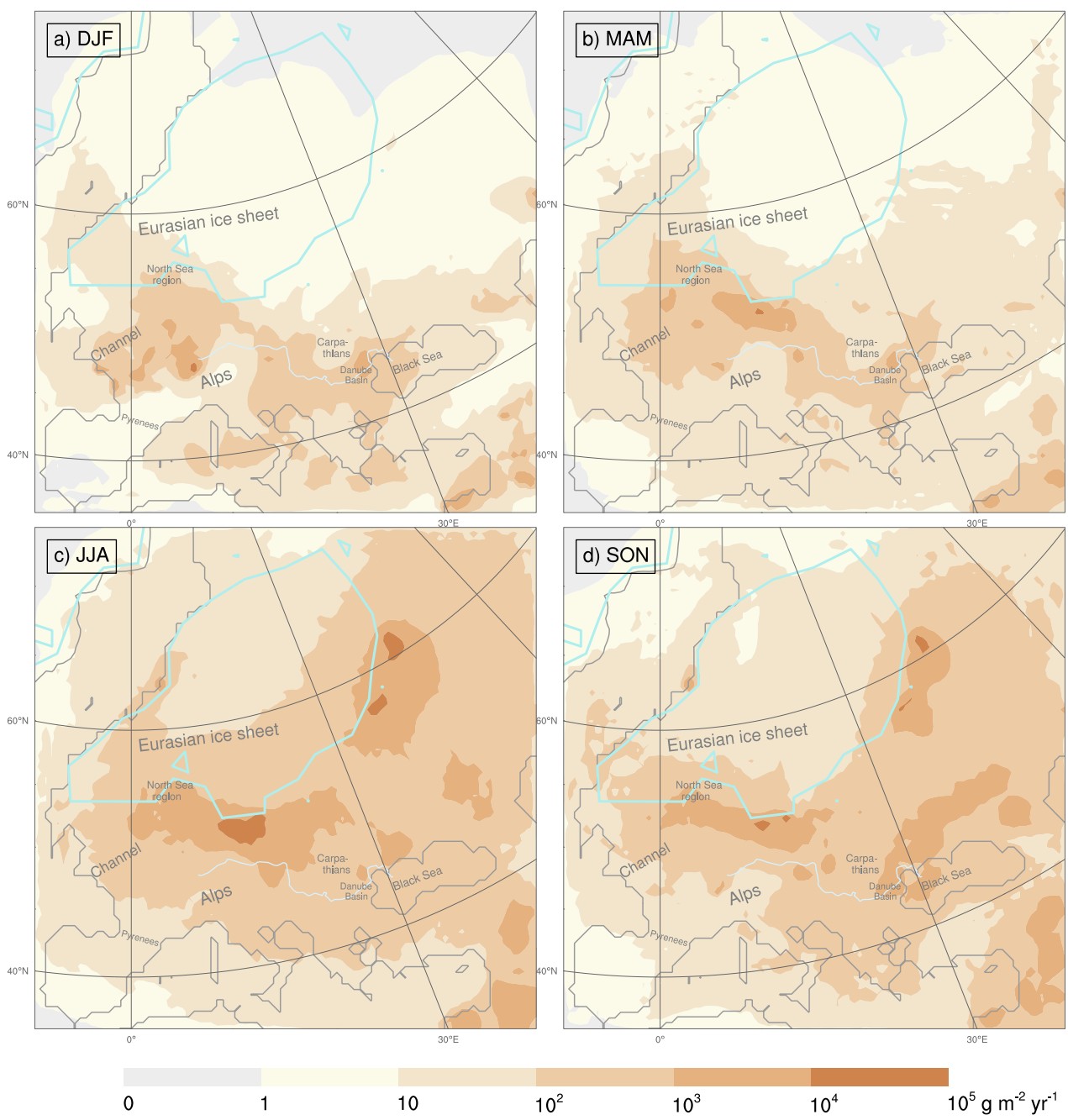

**Figure 6.** Dust deposition rates for a) winter (DJF), b) spring (MAM), c) summer (JJA) and d) autumn (SON) during the Last Glacial Maximum. This reconstruction is based on dynamic downscaling. Ice sheet extents (turquoise; Cline et al., 1984), Danube (light-blue line) and coastlines (grey; Braconnot et al., 2012) are delineated.

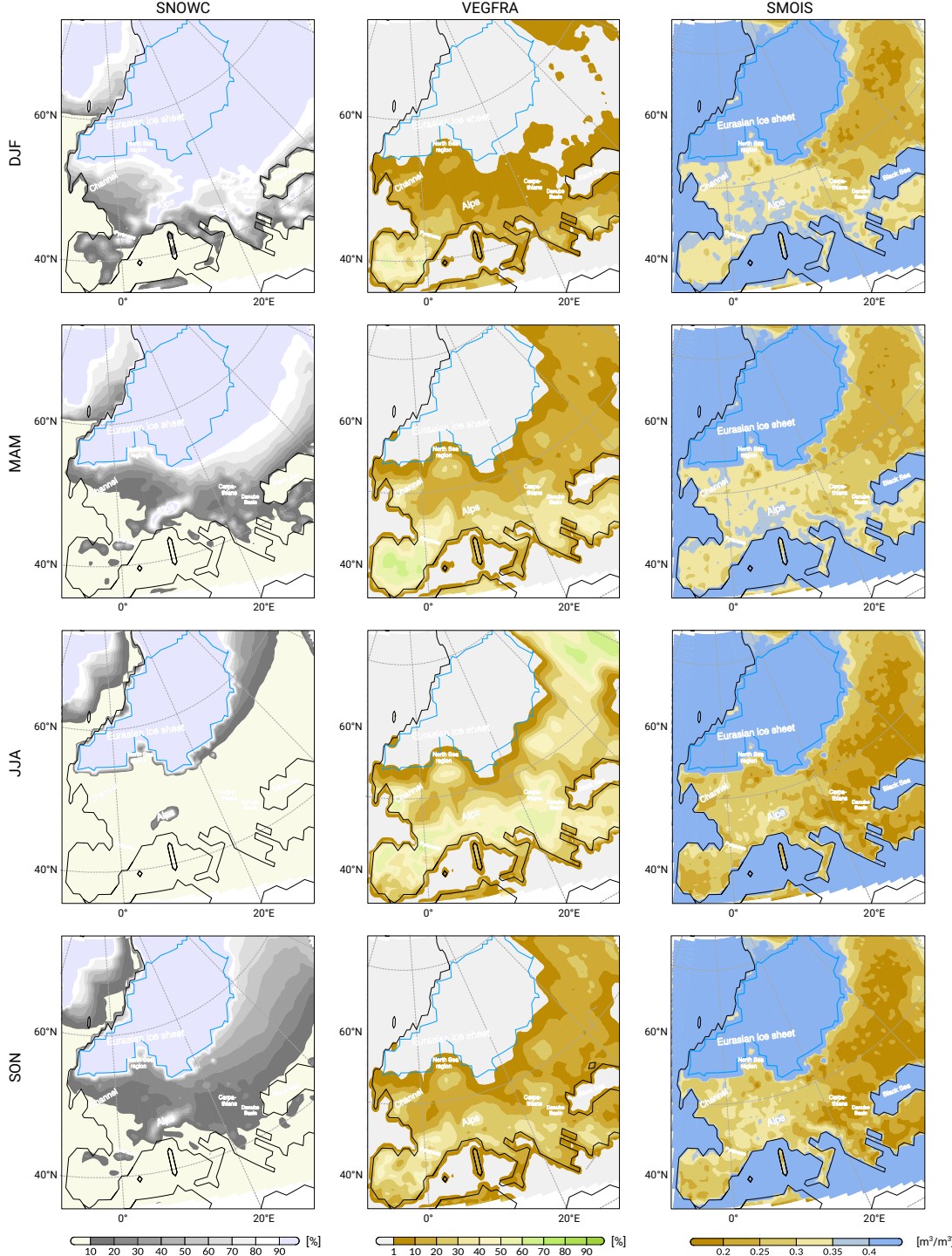

**Figure 7.** Snow cover (%, left column), vegetation cover (%, center) and soil moisture (m$^3$/m$^3$, right), resolved for winter (DJF), spring (MAM), summer (JJA) and fall (SON) for the Last Glacial Maximum. These reconstructions are based on dynamic downscaling.

### 3.6 Wind regime-based dust cycle decomposition

The wind regime occurrence frequency distribution (Table 2) demonstrates the temporal dominance of the east sector winds during the LGM. This temporal dominance likely shaped the dust cycle but the contribution of each wind regime type has so
far not been analyzed. This analysis is provided here by discussing the dust emission and deposition characteristics associated with different CWTs which reveal that the east sector winds caused by far the largest dust emission and depositions during the
LGM (Fig. 8a and 9a). In sum, they generated an average dust emission of 1111 g m$^{-2}$ yr$^{-1}$ (Fig. 8a) which is more than twice of the rate generated by cyclonic regimes (494 g m$^{-2}$ yr$^{-1}$, Fig. 8b). The west sector winds contributed on average even less to
the dust cycle 375 g m$^{-2}$ yr$^{-1}$ (Fig. 8c). Compared to the southerlies (232 g m$^{-2}$ yr$^{-1}$, Fig. 8d), this rate is low for a wind sector that sums the contribution of three wind directions (SW, W, NW).
The cyclonic wind regimes caused the most heterogeneously distributed emissions (Fig. 8b) with four main centers: the largest located in the German-Polish-Czech border region, another in eastern England and the remaining two near the EIS
margin in western Russia. This distribution resembles to a subset of the emission distribution of the east sector winds (Fig. 8a). Together with the location of the CWT reference regions, this resemblance could be explained by the fact that all records
classified as cyclonic must center their cyclonic pressure distribution approximately around the central point for the CWT classification (17.5°E, 47.5°N). This implies that the corresponding emissions could have been triggered by easterlies on the
northern flanks of the cyclones. Dust was hardly emitted from areas on the southern flanks of the cyclones which are commonly affected by fronts and precipitation (Booth et al., 2018). In addition to the dust emission areas that occurred equally during
both regimes (cyclonic and east sector winds), the east sector winds also generated emissions in Austria, Slovakia, Hungary, Ukraine, central Germany, the Danube Basin and the North Sea Basin. In contrast, the west sector winds produced a more
homogeneous distribution of markedly smaller emission rates extending from western Ukraine to the French Atlantic coast. While northwesters with a strong northerly component most likely forced emissions from the German-Polish EIS margin, the
west sector winds and the southerlies controlled the emissions from France, southwestern Germany, the Channel, and the Alps foreland (Fig. 8c and d). The combination of the emission and deposition rate patterns of the east sector winds (Fig. 8a and 9a)
indicates major westwards dust transport along the southern and eastern EIS margin. The conic shape of the deposition rate distribution in western and central Europe (between 10$^2$ and 10$^3$ g m$^{-2}$ yr$^{-1}$) suggests that these depositions can be attributed
to emissions from more eastern sources. The east sector winds also deposited considerable amounts of dust in and south of the Danube Basin as well as along the Danube.
The deposition rates of the cyclonic regimes (Fig. 9b) indicate two main dust transport directions: westwards over central and eastern Europe, whereas southwards over western Europe. More precisely, dust was transported westwards from Poland to
eastern and central Germany, while it was carried southwards from eastern England to the Channel and north-western France up to the Pyrenees foreland. The emission and deposition distributions associated with the west sector winds are almost congruent
(Fig. 8c and 9c). Combining them does not reveal a unique dust transport direction by west sector winds, it rather suggests omnidirectional transports; even a westward transport cannot be excluded e.g. to Scotland, Ireland or areas at the Russian
EIS margin (Fig. 9c). The depositions caused by southerlies show a north-westward transport over central Europe (Fig. 9d).

Considerable amounts of dust (between $10^3$ and $10^5$ g m$^{-2}$ yr$^{-1}$) were transported from sources in western Poland, eastern

Germany and Czechia to northern Germany, Denmark, southern Sweden and the North Sea Basin. The deposition pattern also

suggests a north-westward transport in France.

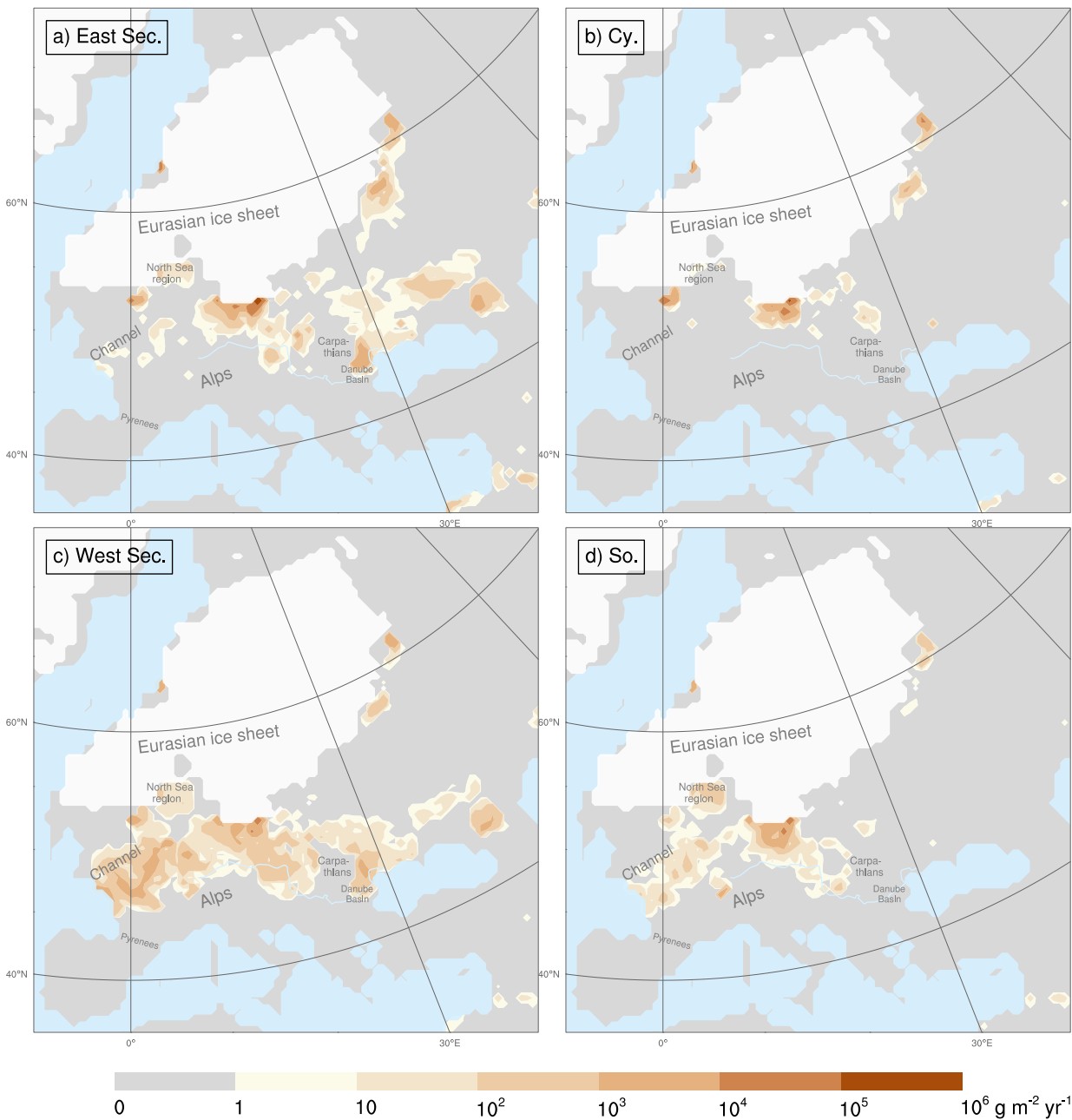

**Figure 8.** Dust emission rate fractions caused by the a) northeasters, easterlies and southeasters, b) cyclonic regimes, c) southwesters, westerlies and northwesters, and d) southerlies during the Last Glacial Maximum. The simulated emission rates are weighted according to the occurrence frequency of the associated wind regime(s) in the Max-Planck-Institute Earth System Model (Table 2). Dust particles up to 20 μm diameter have been considered. The Danube (light-blue line) and the extent of the continental ice sheets (white) are shown.

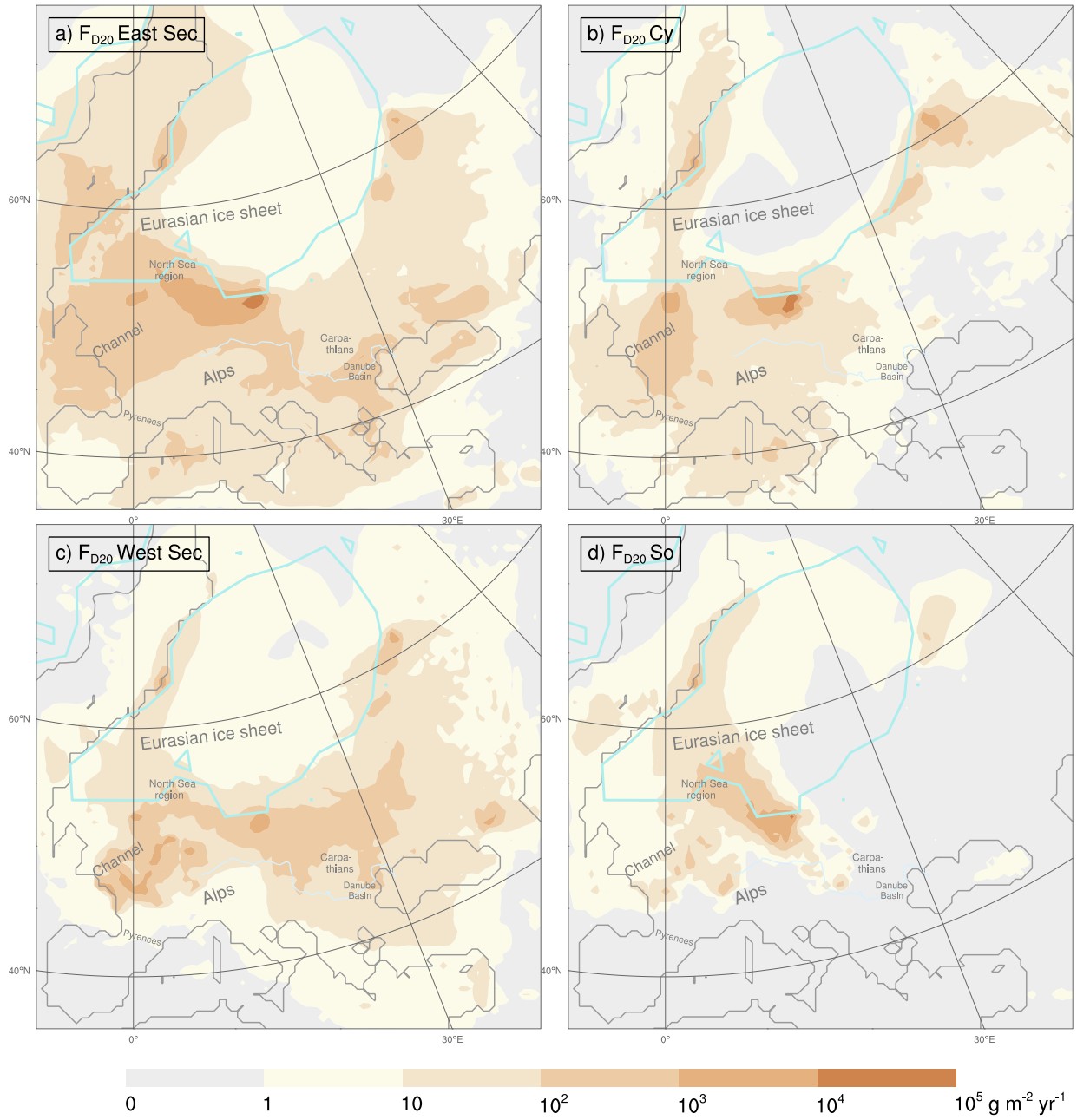

**Figure 9.** Dust deposition rate fractions caused solely by the a) northeasters, easterlies and southeasters, b) cyclonic regimes, c) southwesters, westerlies and northwesters, and d) the southerlies during the Last Glacial Maximum. The simulated deposition rates are weighted according to the occurrence frequency of the associated wind regime(s) in the Max-Planck-Institute Earth System Model (Table 2). Dust particles up to 20 μm diameter have been considered. The ice sheet extents (turquoise; Cline et al., 1984), the Danube (light blue) and the coastlines (grey; Braconnot et al., 2012) are delineated.

# 4 Conclusions

Compared to previous climate-dust model simulations for the LGM, this study presents a dust cycle reconstruction with dust deposition rates that are in much better agreement with the MARs reconstructed from more than 70 different loess deposits across Europe. By taking into account the effect of different wind directions, a more complete understanding of the dust cycle is established. The obtained results corroborate the hypothesis on the linkage between the prevailing dry east sector winds as a major driver of the LGM dust cycle in central and eastern Europe and the loess deposits.

The study demonstrates that the WRF-Chem-LGM model is capable of simulating the glacial dust cycle including emission, transport and deposition. In addition, the suitability of the statistic dynamic approach for regional climate-dust simulations is proven by the similarity of the dynamic and statistic-dynamic downscaling results. In contrast to the dominant present-day westerlies over Europe, the CWT analysis revealed dominant east sector (36%) and cyclonic (22%) wind regimes during the LGM over central Europe. These east sector winds dominated the LGM dust cycle by far during all but the summer season. In summer, they were about as frequent as the cyclonic regimes. The dominance of the east sector winds during the LGM is corroborated by numerous local proxies for the wind and dust transport directions in Europe.

The WRF-Chem-LGM simulations show that almost all dust emission occurred in a corridor that was bounded to the north by the EIS and to the south by the Alps and the Black Sea. Within this corridor, the highest emissions were generated from the dry-fallen flats, the lowlands bordering mountain slopes, and the proglacial areas of the EIS. Most dust was emitted during the summers and autumns of the LGM, probably due to the then vanishing snow cover. The largest dust deposition rates during the LGM occurred near the southernmost margin of the EIS (12–19°E; $10^5$ g m$^{-2}$ yr$^{-1}$), on the North German Plain including adjacent regions and in the southern North Sea region. The agreement between the performed climate-dust simulations for the LGM and the reconstructed MARs from loess deposits corroborates the proposed LGM dust cycle hypothesis.

*Author contributions.* EJS, PL and YS designed the concept of the study. PL performed the dynamic downscaling simulation and created Figure 7. EJS performed the statistic dynamic downscaling, compared the results with the proxy data including the reconstructed loess mass accumulation rates, created the tables and the remaining figures. EJS wrote the paper with contributions from PL and YS.

*Competing interests.* The authors declare that they have no conflict of interest.

*Acknowledgements.* This research was funded by the Deutsche Forschungsgemeinschaft (DFG) through the Collaborative Research Center 806 "Our Way to Europe" (CRC806). P. Ludwig thanks the Helmholtz initiative REKLIM for funding. We thank the German Climate Computing Centre (DKRZ, Hamburg) for providing the MPI-ESM data and computing resources (project 965). We thank the Regional Computing Center (University of Cologne) for providing support and computing time on the high performance computing system CHEOPS.

We thank Qian Xia for preparing model boundary condition data. We thank F. Lehmkuhl, the CRC806 (second phase) members of his group and J. G. Pinto for helpful discussions and comments.

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
