# Peer review of "Linkage between Dust Cycle and Loess of the Last Glacial Maximum in Europe"

_Atmospheric Chemistry and Physics, 2019_

## Referee Comment (RC1) · Anonymous Referee #1 · 23 Oct 2019

Comments to "Linkage between Dust Cycle and Loess of the Last Glacial Maximum in Europe" by Schaffernicht et al.

Manuscript number: acp-2019-693

Quantification of the dust cycle for the Last Glacial Maximum (LGM) is crucial to better understand effects of dust on glacial paleoclimate and paleoenvironments. Loess deposits are paleodust archives providing basic information to test dust cycle models such as the one introduced by Schaffernicht et al. This dust cycle simulation is novel in the sense that it follows a weather typing approach (circulation weather type, CWT, classification) providing deeper insight into regional differences of peak glacial atmospheric circulation in Europe and dust emission/deposition in relation to CWT classes. As demonstrated by the authors simulated bulk and dust MAR values are in good

agreement with the paleodust record (loess MARs) in central Europe, and this study reveals the significant role of easterly and cyclonic wind regimes in LGM dust emission and dust emission/deposition seasonalities (summer/autumn peak). My limited number of (minor) comments/suggestions can be found below as line-by-line comments. This manuscript is recommended for publication in ACP after minor revisions.

Specific comments

Lines 41-46: Bulk and dust MARs should clearly be distinguished in this paragraph, and later in the text. The dust MAR value (100 g/m2/yr) in line 43 is slightly misleading, as this is an estimate of MAR of the <10 micron fraction, so cannot be directly compared to bulk MAR (800 g/m2/yr), as given in the next sentence.

Lines 150-151: Significant loess accumulations are found along the west bank of the Danube river in Hungary, Croatia and Serbia, providing further observational evidence for easterly paleowinds.

Figure 4: Position of the scale is inappropriate as it covers circles representing MAR magnitudes. Also, I suggest adding an x-x plot directly showing a model/paleodata comparison of dust MAR values.

Lines 268 and 278: The dimensions should be g/m2/yr and not kg/m2/yr, I guess.

Lines 297-298: State clearly if this is bulk or dust MAR.

Technical corrections

Line 29: Ujvari et al (2012) is not listed in "References"; or is this the cited study of the authors from 2017? Line 42: Ujvari et al. (2010) cannot be found in the reference list Line 133: missing full stop at the end of sentence Line 249: write "average dust emission"

---

## Referee Comment (RC2) · Anonymous Referee #2 · 2 Nov 2019

This paper examines the contribution of mineral dust cycle to loess deposits in Europe during the Last Glacial Maximum (LGM) using the output from the Max-Plank-Institute Earth System Model (MPI-ESM) and simulations from the WRF-Chem model. The simulated dust deposition rates are largely consistent with site records of mass accumulation rates of the loess deposits. Using statistic dynamical downscaling, it is found that the east sector and cyclonic winds are the dominant circulation regimes during the LGM and thus result in a westward dust transport to the central and eastern Europe. The seasonal variations in dust emission and deposition are also analyzed. Overall, the paper is well organized and written. However, in some places, the purpose of the analysis and methodology need further clarification. My comments are summarized as follows.

[Figure]

Major comments:

1. I'd suggest adding a discussion about the motivation to use the WRF-Chem and MPI-ESM to study dust cycles during the LGM. As mentioned in section 1, results from previous global simulations largely underestimate the mass accumulation rates (MARs) of dust depositions. Is this due to the coarse resolution of the global models or insufficiency of the dust emission schemes to capture certain processes of the dust emission and transport? Or is it related to unrealistic land surface settings for the LGM or misrepresentation of the atmospheric circulation patterns in the models? Similarly, please consider adding explanation/discussion about why current work better captures the magnitude of the MARs in the result section.

2. The purpose of using dynamic downscaling and statistic dynamic downscaling is not quite clear, and the method of dynamic downscaling is somewhat vague. For instance, 30 years of simulations are conducted using dynamic downscaling (line 82). What time period does the simulation cover? Are the 30 years consecutive? What's the setting of sea surface temperature and vegetation cover? More importantly, what's the benefit of using statistic dynamic downscaling? Why not use the results from the dynamic downscaling directly, e.g., by selecting the circulation weather types (CWTs) from the 30-year run?

3. Please consider adding the dust emission scheme (Shao 2004) to section 2, so the readers would have a clearer idea about how dust emission is initiated and constrained in the model. Information such as dust size bins is also needed.

Minor points:

1. By using the CWTs as criteria to conduct statistic dynamic downscaling, it assumes that atmospheric circulation pattern is the dominant factor influencing dust deposition, other factors, such as land surface features (e.g., vegetation coverage, soil moisture), and environmental factors (e.g., wind frequency and magnitude, precipitation) play minor roles. Is this a good assumption? You may want to add some discussion in section

2 about this.

2. Instead of showing schematic of the atmospheric circulation patterns (e.g., Fig. 2), I wonder if you may add figures in text or in the supplement to show the composite of wind patterns either from the MPI-ESM or WRF-Chem simulations to better demonstrate the transport pathways of the dust.

3. Lines 55-56, what's the setting of sea surface temperature for the WRF-Chem? Is it also from the MPI-ESM simulation?

4. Lines 68-69, are the vegetation coverage data monthly or annually?

5. Lines 85- 87, CWT on what level? Near-surface, 850 hPa, or a higher level?

6. Lines 87-88, "to compare the prevailing wind directions over Europe during the Pre-Industrial (PI) and the LGM...". Why not compare with present-day wind direction? In the abstract, "present-day prevailing westerlies" is mentioned, is it referred to the result from the PI simulation?

7. Line 89, what's the difference between the MPI-LGM run and MPI-EMS-P run? How long are these simulations?

8. Line 90, why is this point selected? Is it the center of the Loess?

9. Line 94, what are the differences in 13 simulations? Initial conditions?

10. Line 96, what is the definition of a "CWT set"? 8 consecutive days with the same CWT? Can you please list the number of CWT sets from the MPI-ESM simulation?

11. Fig. 1, can you please add the location of the Loess to the figure?

12. Can you please add some explanation about Table 1? e.g., what's heterogeneous sequence? Why are spin-up records preferred?

13. Line 135, why do you mention Fig. 9a here?

14. Line 181, why do you select 12 and 20 $\mu$m? Is 20 $\mu$m the largest dust size bin in

the model? The results from FD12 in Fig. 4 seem not discussed in section 3.4.

15. Line 214, add "(Fig. 7)" after "vegetation cover".

16. Can you please add the location of dust source (as displayed in Fig. 1) to Fig. 6?

---

## Author Comment (AC1) · 23 Feb 2020

Manuscript number: acp-2019-693

R= Referee#2

A= Authors' reply

R:
This paper examines the contribution of mineral dust cycle to loess deposits in Europe during the Last Glacial Maximum (LGM) using the output from the Max-Plank-Institute Earth System Model (MPI-ESM) and simulations from the WRF-Chem model. The simulated dust deposition rates are largely consistent with site records of mass accumulation rates of the loess deposits. Using statistic dynamical downscaling, it is found that the east sector and cyclonic winds are the dominant circulation regimes during the LGM and thus result in a westward dust transport to the central and eastern Europe. The seasonal variations in dust emission and deposition are also analysed. Overall, the paper is well organized and written. However, in some places, the purpose of the analysis and methodology need further clarification.

A:
We thank the referee very much for the valuable comments and suggestions to improve the manuscript. Subsequently, the referee's comments are addressed point by point.  It is our aim to fulfil the demands to publish this manuscript  in ACP.

(Reference keys that are not fully written out in this document refer to the References section of the updated manuscript)

**Major comments**

R2.1    I'd suggest adding a discussion about the motivation to use the WRF-Chem and MPI-ESM to study dust cycles during the LGM.

A       The MPI-ESM was used as its 1850–2005 experiment reproduces best the recent observed wind distribution over central Europe. This result was found by comparing the CWT distribution of four different global earth system/circulation models (MPI-ESM, CCSM, MRI and MIROC) to reanalyses data for central Europe  (Schaffernicht, Erik Jan:  *Linkage between Dust Cycle and European Loess in the Last Glacial Maximum Determined by Atmospheric Model Simulations.*
Inaugural Dissertation, PhD thesis, University of Cologne, Germany, 2018.
https://kups.ub.uni-koeln.de/9036/
http://kups.ub.uni-koeln.de/id/eprint/9036
).

In addition, access to boundary conditions that are updated frequently enough to carry out the intended WRF-Chem-LGM experiments was only offered by the MPI-LGM.

The WRF-Chem was chosen to be the core for the LGM dust simulation model because it has already been evaluated successfully in numerous recent studies comparing its dust simulations with observations (Bian et al. 2011, Zhao et al. 2011, Zhao et al. 2012, Rizza et al. 2016, Baumann el al. 2019).

R2.2    As mentioned in section 1, results from previous global simulations largely underestimate the mass
        accumulation rates (MARs) of dust depositions. Is this due to the coarse resolution of the global
        models...
A       Yes and due to local small scale dust sources and deposition processes (Werner et al. 2002).

R2.3    ...or insufficiency of the dust emission schemes to capture certain processes of the dust emission and
        transport?....
A       Yes (Werner et al. 2002) and also a missing process or a low sensitivity in the dust model is possible
        (Hopcroft et al. 2015 JGRA).

R2.4    ...Or is it related to unrealistic land surface settings for the LGM...
A       Yes: Probably due to the missing glaciogenic dust sources (Mahowald et al. 2006, Hopcroft et al
        2015JGRA) and parameterizations of source regions and source material availability are
        undersensitive to LGM conditions (Hopcroft et al. 2015 JGRA)

R2.5    ...or misrepresentation of the atmospheric circulation patterns in the models?
A       Yes, e.g. Ludwig et al. (2016).  Lacking interannual variability and dust storm events might be another
        factor (Hopcroft et al. 2015 JGRA).
        Corresponding statements have been added to the manuscript.

R2.6    Similarly, please consider adding explanation/discussion about why current work better captures the
        magnitude of the MARs in the result section.

A       Corresponding explanations/discussion has been added to the manuscript in the section:
        "Conforming Dust Deposition and Loess Accumulation Rates".

        Current work captures better the magnitude of the MARs because:

        - the regional simulations are run with a much higher resolution compared to previous studies
        - its simulations include additional dust sources that likely existed due to the glacial topography.
        - it takes into account Ginoux's dust function (Ginoux et al. 2001) and resolves Europe at higher
        spatiotemporal resolution
        - it takes into account dynamic soil moisture, vegetation and snow cover
        - its boundaries are driven by the LGM simulation of the MPI-ESM. For the end of the 20th century,
        this ESM reproduced the observed atmospheric circulation over Europe better than other ESM/GCMs
        (Ludwig et al. 2016).
        - it uses a well-tested and observation-proofed dust emission scheme (Shao 2004)

R2.7    The purpose of using dynamic downscaling and statistic dynamic downscaling is not quite clear, and
        the method of dynamic downscaling is somewhat vague. For instance, 30 years of simulations are
        conducted using dynamic downscaling (line 82). What time period does the simulation cover? Are the
        30 years consecutive?

A        The dynamic and statistic dynamic downscaling serve to simulate the glacial dust cycle at high resolution using the WRF-Chem-LGM including seasonal and circulation weather type dependent aspects. More details are provided in the answer to R2.11.

The MPI-LGM (simulation in equilibrium setup) covers average LGM conditions. Its arbitrary timestamp is 1919-01-01 to 1948-12-31.

The 30 years are consecutive (see line 83 in the initial manuscript).

R2.8     What's the setting of sea surface temperature?

A        The sea surface temperature and sea ice cover are updated daily based on the corresponding MPI-LGM variables.

R2.9     What's the setting of vegetation cover?

A        The vegetation cover has been reconstructed from the CLIMAP LGM maximum vegetation cover and the vegetation dynamics extracted from the present-day WRF geo data. Details can be found in manuscript line 67 and Supplementary Table S2 and S3.

R2.10    More importantly, what's the benefit of using statistic dynamic downscaling?

A        A high resolved (i.e. approximately 50 km grid spacing) reconstruction of the glacial dust cycle based on statistic dynamic downscaling requires much less computation time. It is the first proof-of-concept that statistic dynamic downscaling not only works for wind regime analyses but also for reconstructing the mineral dust cycle.  Also, statistic dynamic downscaling enables analysing the dust cycle by wind regimes.

R2.11    Why not use the results from the dynamic downscaling directly, e.g., by selecting the circulation weather types (CWTs) from the 30-year run?

A        Extracting  CWT samples directly from the 30-year run would imply including a non-quantifiable amount of background dust; thus, the deposition rates extracted in this case might not solely be related to the specific CWT.
Also, the results based on the statistic dynamic downscaling of the 100-year MPI-LGM are intended for comparison to the dynamic downscaling results that base on only 30 years. Selecting CWTs only from the 30-year run would ignore 70 years of daily records for the LGM wind field over Europe.
In addition, when designing the concept for this study, there was a lack of studies showing that a dynamic climate-dust cycle downscaling over 30 years is possible within the numerical limitation of the available high performance computer (HPC).  As the implementation of the statistical dynamic downscaling implied a reduction of the required simulation days by a factor of ten without missing any major wind direction feature, it is a promising approach. This finding can be important in particular  for larger domains and/or models requiring  much more computation time for dynamic downscaling.

R2.12    3. Please consider adding the dust emission scheme (Shao 2004) to section 2, so the readers would have a clearer idea about how dust emission is initiated and constrained in the model. Information such as dust size bins is also needed.

A        The 0–20 μm particle size range is partitioned in five dust size bins: 0-2, 2-3.6, 3.6-6, 6-12, 12-20 μm.
         Added to the manuscript in the first paragraph of the section "Data and Methods".

         The dust emission scheme (Shao 2004) is referred to in Section 2, first paragraph (e.g., in manuscript
         line 60): "This mode implies the application of the size-resolved University of Cologne dust emission
         scheme (Shao2004) […]". The structure and implementation of this dust emission scheme is extensive
         and cannot be summarized in a few sentences. It consists of many physical, mathematical and
         numerical/technical aspects, which are discussed in detail in 'Shao (2004)'. If there is a specific detail
         of the emission scheme that Referee2 requires here, we kindly ask Referee2 to let us know which
         particular formula or detail (s)he misses here. It would go beyond the scope of this study to
         (re-)discuss 'Shao (2004)'. It is kindly recommended to read 'Shao (2004)' for more information on
         the dust emission scheme.

**Minor points**

R2.13    1. By using the CWTs as criteria to conduct statistic dynamic downscaling, it assumes that atmospheric
         circulation pattern is the dominant factor influencing dust deposition, other factors, such as land
         surface features (e.g. vegetation coverage, soil moisture), and environmental factors (e.g. wind
         frequency and magnitude, precipitation) play minor roles. Is this a good assumption? You may want to
         add some discussion in section 2 about this.

A        The deposition location(s) depend(s) on different factors, foremost the speed and direction (and thus
         implicitly also the emission locations) of the wind that caused emission and transport of the dust
         particles (Darmenova et al. 2009). This constrains the potential locations for deposition. In addition,
         land surface features can affect the deposition, yet, it is beyond the scope of this study to quantify their
         proportional effect in the WRF-Chem-LGM among other factors such as the CWTs.

         The atmospheric circulation patterns are the most relevant factor for the complete dust cycle including
         emission, transport and deposition. They include environmental factors implicitly, e.g. the wind regime
         frequency and precipitation likelihood. The effects of soil moisture, snow and vegetation cover on the
         dust cycle emerge in particular on the seasonal scale (Fig. 9) and are taken into account by the applied
         dust emission scheme. The agreement between the statistic dynamic and the dynamic downscaling
         results demonstrates that the CWT-focused approach captures the dominant factors well. (Manuscript
         updated)

R2.14    2. Instead of showing schematic of the atmospheric circulation patterns (e.g., Fig. 2), I wonder if you
         may add figures in text or in the supplement to show the composite of wind patterns either from the
         MPI-ESM or WRF-Chem simulations to better demonstrate the transport pathways of the dust.

A        The wind patterns are shown by the grey lines in Fig. 2. They were extracted directly from the WRF-
         Chem-LGM experiments. (The caption of Fig. 2 has been updated accordingly.)

         The composite of wind patterns over Europe from the MPI-ESM can be found and has already been
         discussed in Ludwig et al. (2016).

R2.15    Lines 55-56, what's the setting of sea surface temperature for the WRF-Chem? Is it also from the MPI-ESM simulation?

A        Yes, it is from the MPI-LGM.

R2.16    4. Lines 68-69, are the vegetation coverage data monthly or annually?
A        They are monthly:
         "...using the corresponding **monthly** fractions of the present-day WRF maximum vegetation cover..." (near line 69).
         Based on the CLIMAP maximum LGM vegetation cover reconstruction, a monthly vegetation cover was calculated as an analog to the present-day monthly WRF vegetation dynamics

R2.17    5. Lines 85-87, CWT on what level? Near-surface, 850 hPa, or a higher level?
A        CWTs are based on the mean sea level pressure; references Jones et al 1993, Jones et al 2003 are in the manuscript and provide the details (near line 85).

R2.18    6. Lines 87-88, "to compare the prevailing wind directions over Europe during the Pre-Industrial (PI) and the LGM...". Why not compare with present-day wind direction? In the abstract, "present-day prevailing westerlies" is mentioned, is it referred to the result from the PI simulation?

A        The LGM CWT  frequencies have also been compared with present-day CWT frequencies:

         Table 2-extended:

|  | C | A | NE | E | SE | S | SW | W | NW | N |
|---|---|---|---|---|---|---|---|---|---|---|
| LGM | 22.2 | 8.9 | 12.4 | 13.4 | 10.2 | 9.7 | 6.8 | 4.3 | 5.0 | 7.0 |
| PI | 10.6 | 24.1 | 7.9 | 5.2 | 4.9 | 7.6 | 11.6 | 11.1 | 9.4 | 8.3 |
| present-day | 10.6 | 23.9 | 7.3 | 5.1 | 4.7 | 7.5 | 12.4 | 11.4 | 9.2 | 8.0 |

          The present-day frequencies are not shown in Table 2  for consistency. The LGM MPI-ESM experiment is an equilibrium experiment (without transient forcing). Thus, comparing it to equilibrium experiments such as for the PI is more robust because this does not induce uncertainty due to different kind of permanency of model forcing setups. Table 2-extended shows that the  corresponding present-day CWT frequencies obtained from the transient MPI-ESM experiment are indeed almost identical with the PI CWT frequencies.

         The manuscript is updated in section "East Sector Winds and Cyclones over Central Europe" by:

         The CWT frequencies for the present (not shown) and PI are very similar, therefore it is possible to use the term present-day to refer to both the PI and the actual present-day frequencies.

R2.19    7. Line 89, what's the difference between the MPI-LGM run and MPI-ESM-P run? How long are these simulations?

A        The MPI-ESM-P is a typo. The sentence in line 89 (Data and Methods, paragraph 4) is now updated. The MPI-LGM is 100 years long.

R2.20   8. Line 90, why is this point selected? Is it the center of the Loess?

A       Yes, approximately; taking into account that the present-day loess sites exist for example in northwestern France and also in Ukraine. Also, it (17.5°E, 47.5°N) is located near the Carpathian basin which is a prominent loess region.

R2.21   9. Line 94, what are the differences in 13 simulations? Initial conditions?

A       Yes, the 13 simulations differ slightly in their initial and boundary conditions.

R2.22   10. Line 96, what is the definition of a "CWT set"? 8 consecutive days with the same CWT? Can you please list the number of CWT sets from the MPI-ESM simulation?

A       A CWT set consists of 8 consecutive days. At least the CWT of the third, forth and fifth day of these 8 consecutive days (also called "the 3 main days", cf. Table 1) must be identical to the requested CWT. For most of the 13 simulations, all of their eight consecutive days belong to the same CWT.

        130 CWT sets have been chosen from the MPI-LGM. (cf. near line 94)

R2.23   11. Fig. 1, can you please add the location of the Loess to the figure?

A       Fig. 1 focuses on the numerical WRF-Chem-LGM setup (i.e. the model input data) and it contains already the dotted area that shows that all areas outside this area are excluded from being a potential dust source. It would overload this map with too many colours, shades, lines and symbols if we add more independent data sets that need to be clearly distinguishable without affecting the already shown layers. Adding these loess locations would also suggest that they were used as input data for the WRF-Chem-LGM experiments. This would be misleading because the loess sites samples are only used to compare the WRF-Chem-LGM results to. They are not part of any simulation.

R2.24   12. Can you please add some explanation about Table 1? e.g., what's heterogeneous sequence? Why are spin-up records preferred?

A       Table 1:
        Each CWT sequence consists of 8 consecutive days, i.e. 2 spin-up, 3 main and 3 tracking days. In all CWT sequences, the CWT of the 3 main days is identical and defines the desired CWT for the whole 8-day episode.
        For rare cases for which no 13 distinct 8-day samples for the same CWT were available in the MPI-LGM, the strict selection criteria has been weakened. That is, the 2 spin-up days are then allowed to be samples of a different CWT. If applying this weakened selection criteria was still insufficient to extract 13 different 8-day episodes from the MPI-LGM, then the selection criteria was weakened further, that is, also the 3 tracking days are allowed to deviate from the CWT of the main days.

        This approach implies that the priority of the 3 tracking days to fit to the CWT of the main days is higher (++) than that of the 2 spin-up days (+).

        An episode is called *heterogeneous* if at least 1 (out of 5 possible) record differs from the desired CWT of the episode's main days.

        The caption of Table1 has been updated.

R2.25    13. Line 135, why do you mention Fig. 9a here?
A        Fig. 9a is mentioned here since it shows the importance of east sector winds.

R2.26    14. Line 181, why do you select 12 and 20 μm?

         Is 20 μm the largest dust size bin in the model?

         The results from FD12 in Fig. 4 seem not discussed in section 3.4.

A        20 μm is selected because it is the upper limit of particle sizes included in the WRF-Chem-LGM with the
         applied dust emission scheme.

         12 μm is selected because the WRF-Chem-LGM particle size bin distribution provides this limit as the
         nearest approximation of the deposition values for 10 μm particle size limit that are commonly
         published in studies based on mass accumulation rates (MAR10 = MAR 10 μm) from loess fieldwork
         samples.
         Yes, 20 μm is the largest dust size bin in the model.  (Manuscript updated)

R2.27    15. Line 214, add "(Fig. 7)" after "vegetation cover".
A        Thanks for this comment. In the new version, this is corrected. We assume Referee2 refers to line 213
         as "veg. cover" does not occur in line 214.

R2.28    16. Can you please add the location of dust source (as displayed in Fig. 1) to Fig. 6?

A        Parts of Fig. 1 and Fig. 6 are possibly misunderstood:
         Fig. 1 does not show any dust source location.  Instead,  the dotted area limits the area for **potential**
         dust sources. That is, it rather excludes regions (i.e. the non-dotted) as potential dust sources since they
         have a very low or zero potential erodibility according to Ginoux's time-independent dust function
         (Ginoux et al. 2001). If the dotted regions eventually become dust sources depends on additional
         dynamic factors such as e.g. the atmospheric circulation (in particular the wind speed) and surface
         conditions (e.g. snow and vegetation cover, soil moisture). As Fig. 6 displays seasonal dust cycle
         aspects, it should be noted, that (in addition to the before mentioned) the dotted area results from
         spatiotemporal average (i.e. annual) LGM conditions.  It does not take into account (nor adapt to) the
         seasonal changes that occurred during the LGM.

         Fig. 6 focuses on the seasonal dust deposition rates. Adding Ginoux's time-independent potential
         erodibility mask (shown in Fig. 1) would suggest that this mask is adjusted to seasonal differences and
         particularly appropriate for seasons.  Yet, this suggestion is misleading and would create a misleading
         understanding of the Fig.6 for the reader.

         Nevertheless, the referee's request is already met in Fig. 5, which shows the location of the dust
         sources resolved by season. In addition, Fig. 5 shows the seasonal dust emission rates of the dust
         sources.

         For clarity and readability, it is suggested to **not** add further colours, shapes, contours nor shades
          to Fig. 6.  Otherwise, it would become overloaded.

---

## Author Comment (AC2) · 23 Feb 2020

Comments to "Linkage between Dust Cycle and Loess of the Last Glacial Maximum in  Europe" by Schaffernicht et al.

Manuscript number: acp-2019-693

R= Referee1

A= Authors' reply

R:
Quantification of the dust cycle for the Last Glacial Maximum (LGM) is crucial to better understand effects of dust on glacial paleoclimate and paleoenvironments. Loess deposits are paleodust archives providing basic information to test dust cycle models such as the one introduced by Schaffernicht et al. This dust cycle simulation is novel in the sense that it follows a weather typing approach (circulation weather type, CWT, classification) providing deeper insight into regional differences of peak glacial atmospheric circulation in Europe and dust emission/deposition in relation to CWT classes. As demonstrated by the authors simulated bulk and dust MAR values are in good agreement with the paleodust record (loess MARs) in central Europe, and this study reveals the significant role of easterly and cyclonic wind regimes in LGM dust emission and dust emission/deposition seasonalities (summer/autumn peak). My limited number of (minor) comments/suggestions can be found below as line-by-line comments. This manuscript is recommended for publication in ACP after minor revisions.

A:
We thank Referee1 very much for the suggestions and comments on our manuscript as they contribute to improving the submitted manuscript. Our point by point answers to Referee1's comments follow.

**Specific comments**

R1.1  Lines 41-46: Bulk and dust MARs should clearly be distinguished in this paragraph, and later in the text.

A      These terms are consistently used in the complete manuscript:
1) "MAR":
MAR is equivalent to "bulk MAR".  It refers only to fieldwork-based reconstructed accumulations rates without any limitation of particle size.

2) "MAR10":
MAR10 refers only to fieldwork-based reconstructed accumulation rates of particles up to 10 micron diameter.

3) "dust deposition rates":
This term refers to only any kind of numerical-model simulated deposition rate without limiting or specifying its particle size range. For example, the particle sizes range in the WRF-Chem-LGM includes particles up to 20 micron.

4a) "$F_{D20}$":
It labels the WRF-Chem-LGM simulated deposition rates up to 20 micron particle size.

4b) "$F_{D12}$":
It labels the WRF-Chem LGM simulated deposition rates up to 12 micron particle size.

4c) "$F_D$":
It refers to $F_{D20}$ and $F_{D12}$.

R1.2   The dust MAR value (100 g/m2/yr) in line 43 is slightly misleading, as this is an estimate of MAR of the <10 micron fraction, so cannot be directly compared to bulk MAR (800 g/m2/yr), as given in the next sentence.

A       Referee1 claims that the 100g/m2/yr (i.e. the upper limit of the numerically-simulated deposition rates of the cited studies) is an estimate based only on MAR10 (= "MAR of the < 10 micron fraction").

This is not the case, i.e.:  The 100g/m2/yr are no estimate of the MAR10.

Reasoning:
Only two of the five references that we cite to prove our statement are based on simulating particles < 10micron. In one of the remaining three references, for example, the particle size is limited by 1000micron.

In summary, the 100 g/m2/yr bases on at least five published simulation results, all of which differ from one another in their particle size ranges. This upper deposition rate limit (= 100 g/m2/yr) can thus not be related to a common particle size range. At least two studies explicitly use a particle size range that exceeds 10 microns.

Thus, our statement  that the numerical simulations published up to now significantly underestimate the field research-based reconstructed accumulation rates for the LGM,   remains valid.

R1.3   Lines 150-151:  Significant loess accumulations are found along the west bank of the Danube river in Hungary, Croatia and Serbia, providing further observational evidence for easterly paleowinds.

A       Referee1's statement is very much appreciated. If Referee1 provides us with a full reference to a peer-reviewed article confirming this, we will be happy to include this loess-related statement in the manuscript.

R1.4   Figure 4:
a) Position of the scale is inappropriate as it covers circles representing MAR magnitudes.

A       The position of colour bar was intentionally set like this to provide more space for the important content, i.e. the panels showing the maps. This design was chosen because the order of magnitude of the MARs are represented solely by the diameter of their circles. It is therefore sufficient to display only two thirds of a full circle line to uniquely define and recognize its diameter.
However, if Referee1 (after reading this reasoning) continues to insist that all circles are shown completely above the colour bar (which means that the main content of the figure is reduced), we will comply to his/her request.

R1.5   b) Also, I suggest adding an x-x plot directly showing a model/paleodata comparison of dust MAR values.

A       Such an x-x plot (R1.5-Fig. 1) showing the WRF-Chem-LGM-based dust deposition rates compared to the MARs can be found below.

The x-x plot shows that the WRF-Chem-LGM based dust deposition rates are in good agreement with the fieldwork-based reconstructed MARs. Ultimately, it must be taken into account that the reconstructed MARs for certain areas show great local variability, the cause of which is probably due to conditions that can still not be completely resolved in the applied WRF-Chem-LGM grid. It is therefore possible that some small-scale features cannot be reproduced in the grid resolution of this study. In addition, the small-scale land surface conditions in Europe during the LGM are so far not sufficiently known nor researched.

[Figure]

Figure 1

R1.6  Lines 268 and 278: The dimensions should be g/m2/yr and not kg/m2/yr, I guess.
A      Thanks for this comment. In the new version, this is corrected.

R1.7  Lines 297-298: State clearly if this is bulk or dust MAR.

A      "The largest dust deposition rates during the LGM occurred […]" refers to the WRF-Chem-LGM simulations. Any other wording would be inconsistent with all other sections of the manuscript. A further distinction between WRF-Chem-LGM particles up to 12 microns and up to 20 microns is not necessary here, since this sentence applies to both deposition rates (based on 12 and 20 micron). If we had referred to results from fieldwork, we would have used the terms MAR (or MAR10) instead.

Technical corrections

R1.8    Line 29: Ujvari et al (2012) is not listed in "References"; is this the cited study of the authors from 2017?
A       Due to a UTF-8 sorting error, it is listed at the end of the References list.  Thanks for pointing to this. In the new version, the References list is re-sorted.

R1.9    Line 42: Ujvari et al. (2010) cannot be found in the reference list
A       Thanks for pointing to this; in the new version, this is corrected.

R1.10   Line 133: missing full stop at the end of sentence
A       Thanks for pointing to this; in the new version, this is corrected.

R1.11   Line 249: write "average dust emission"
A       Thanks for pointing to this; in the new version, this is corrected.